# Agnostic Learning under Targeted Poisoning: Optimal Rates and the Role of Randomness

**Bogdan Chornomaz**[*]
Department of Mathematics
Technion Israel Institute of Technology
Haifa, Israel
markyz.karabas@gmail.com

**Yonatan Koren**
Department of Mathematics
Technion Israel Institute of Technology
Haifa, Israel
yonatankoren@campus.technion.ac.il

**Shay Moran** [†]
Departments of Mathematics, Computer Science, and Data and Decision Sciences
Technion Israel Institute of Technology and Google Research
Haifa, Israel
smoran@technion.ac.il

**Tom Waknine**[‡]
Department of Mathematics
Technion Israel Institute of Technology
Haifa, Israel
tom.waknine@campus.technion.ac.il

## Abstract

We study the problem of learning in the presence of an adversary that can corrupt an $\eta$ fraction of the training examples with the goal of causing failure on a specific test point. In the realizable setting, prior work established that the optimal error under such instance-targeted poisoning attacks scales as $\Theta(d\eta)$, where $d$ is the VC dimension of the hypothesis class [Hanneke, Karbasi, Mahmoody, Mehalel, and Moran (NeurIPS 2022)]. In this work, we resolve the corresponding question in the agnostic setting. We show that the optimal excess error is $\widetilde{\Theta}(\sqrt{d\eta})$, answering one of the main open problems left by Hanneke et al. To achieve this rate, it is necessary to use randomized learners: Hanneke et al. showed that deterministic learners can be forced to suffer error close to $1$ even under small amounts of poisoning. Perhaps surprisingly, our upper bound remains valid even when the learner's random bits are fully visible to the adversary. In the other direction, our lower bound is stronger than standard PAC-style bounds: instead of tailoring a hard distribution separately for each sample size, we exhibit a single fixed distribution under which the adversary can enforce an excess error of $\Omega(\sqrt{d\eta})$ infinitely often.

[*]Bogdan Chornomaz is supported by the European Union (ERC, GENERALIZATION, 101039692).

[†]Robert J. Shillman Fellow; supported by ISF grant 1225/20, by BSF grant 2018385, by Israel PBC-VATAT, by the Technion Center for Machine Learning and Intelligent Systems (MLIS), and by the European Union (ERC, GENERALIZATION, 101039692). Views and opinions expressed are however those of the author(s) only and do not necessarily reflect those of the European Union or the European Research Council Executive Agency. Neither the European Union nor the granting authority can be held responsible for them.

[‡]Tom Waknine is supported by the European Union (ERC, GENERALIZATION, 101039692) and by the ISF grant.

39th Conference on Neural Information Processing Systems (NeurIPS 2025).

# 1 Introduction

Imagine a social network like Facebook or X (formerly Twitter), where recommendation algorithms curate the content shown to users. A business or political entity might attempt to manipulate the system by injecting carefully crafted fake interactions — likes, shares, and comments — into the training data, aiming to subtly boost the visibility of its content for specific target users. The goal is not to disrupt overall system performance, but to force an error on particular instances of interest.

Such scenarios fall outside the scope of classical models of learning, such as PAC learning, which assume that the training data and the test example are independent. When an adversary can modify the training data based on knowledge of the test instance, this independence assumption breaks down. This motivates the study of targeted data poisoning attacks, where the adversary corrupts part of the training set in order to cause failure on a specific test point.

## 1.1 Main Results

In this subsection, we present the learning model and our main results. The presentation here assumes familiarity with standard concepts from classical learning theory; a more self-contained exposition will be provided in Section 4.

We focus on the instance-targeted poisoning model (see, e.g., Barreno, Nelson, Sears, Joseph, and Tygar [2006]), parameterized by a poisoning budget $\eta \in (0,1)$ and a sample size $n > 0$. The interaction proceeds as follows:

---

**Learning under Instance-Targeted Poisoning**

1. The adversary selects a distribution $\mathcal{D}$ over labeled examples $(x, y) \in \mathcal{X} \times \{0, 1\}$.

2. A training sample $S$ of $n$ examples and a target example $(x, y)$ are drawn independently from $\mathcal{D}$.

3. The adversary observes both $S$ and $(x, y)$, and modifies up to an $\eta$-fraction of the examples in $S$ to produce a poisoned sample $S'$.

4. The learner receives $S'$ and outputs a hypothesis $\mathcal{A}(S') : \mathcal{X} \to \{0, 1\}$.

5. The learner succeeds if $\mathcal{A}(S')(x) = y$ and fails otherwise.

---

We study this model with respect to a fixed hypothesis class $\mathcal{H} \subseteq \{0, 1\}^{\mathcal{X}}$. As in classical learning theory, we distinguish between the realizable and agnostic settings:

- In the *realizable* case, there exists a hypothesis $h \in \mathcal{H}$ with zero loss on $\mathcal{D}$.

- In the *agnostic* case, no assumption is made on $\mathcal{D}$, and the learner aims to compete with the minimal achievable loss over $\mathcal{H}$.

Previous work by Gao, Karbasi, and Mahmoody [2021a] showed that if the poisoning budget $\eta$ vanishes with the sample size ($\eta = o(1)$ as $n \to \infty$), then PAC learnability of $\mathcal{H}$ implies learnability under instance-targeted poisoning, both in the realizable and agnostic settings. Their algorithms, however, relies crucially on $\eta$ being negligible. Subsequently, Hanneke, Karbasi, Mahmoody, Mehalel, and Moran [2022] studied the problem for general values of $\eta$, and provided tight bounds on the achievable error in the realizable setting as a function of $\eta$ and the VC dimension $\mathrm{VC}(\mathcal{H})$. They showed that an error rate of $O(\eta \cdot \mathrm{VC}(\mathcal{H}))$ is achievable, and that this rate is optimal up to constant factors. Remarkably, this optimal rate can already be attained by deterministic learners. In the agnostic setting, however, Hanneke et al. [2022] revealed a strikingly different phenomenon: deterministic learning under instance-targeted poisoning is impossible. Specifically, they showed that for any deterministic learner, an adversary can drive the learner's error close to 1, even when $\eta$ is small (e.g., $\eta = O(1/\sqrt{n})$). This naturally raises the following question:

**Main Question**
Is agnostic learning under instance-targeted poisoning possible using randomized learners?
What is the optimal achievable excess error?

We show that randomness indeed circumvents the impossibility result established by Hanneke et al. [2022] for deterministic learners. Furthermore, we obtain nearly tight bounds on the optimal achievable excess error, matching up to logarithmic factors.

> **Theorem 1** (Main Result). *Let $\mathcal{H}$ be a concept class with VC dimension $d$, and let $\eta \in (0,1)$ be the poisoning budget. Then, the following hold:*
>
> - *(Upper Bound) There exists a randomized learning rule such that for any distribution and against any adversary with poisoning budget $\eta$, and sample size of at least $n \geq \frac{1}{\eta}$, the excess error is at most $\tilde{O}\left(\sqrt{d\eta}\right)$.[1]*
>
> - *(Lower Bound) For every learning rule, and $n > 0$, there exist a distribution $\mathcal{D}$ and an adversary with poisoning budget $\eta$ such that the excess error is at least $\tilde{\Omega}\left(\min\left\{\sqrt{d\eta}, 1\right\}\right)$.*
>
> *In particular, for every sufficiently large sample size $n$, the optimal achievable excess error under instance-targeted poisoning is*
>
> $$\tilde{\Theta}\left(\min\{\sqrt{d\eta}, 1\}\right).$$
>
> ---
>
> [1]Note that if $n < \frac{1}{\eta}$, then the adversary's budget is too small to corrupt any examples. In this case, the problem reduces to standard PAC learning, for which the optimal error rate is $\Theta(\sqrt{d/n})$.

The upper bound in Theorem 1 and its quantitative version, Theorem 4, are achieved by a *proper* randomized learner, which selects its output hypothesis with probability proportional to the exponential of (minus) its empirical loss — in the spirit of multiplicative weights algorithms and the exponential mechanism from differential privacy. The connection between data poisoning and differential privacy [Dwork, McSherry, Nissim, and Smith, 2006] is intuitive: differential privacy requires that an algorithm's output be stable under adversarial changes to a single training example. This guarantee extends—with some quantitative degradation—to *group privacy*, which ensures stability under changes to multiple examples. This notion is closely related to the goal in our setting, where the adversary may corrupt an $\eta$-fraction of the training data.

The lower bound is established by first analyzing a simpler setting, the *poisoned coin guessing problem*, and then using a kind of direct-sum argument to extend the bound to arbitrary VC classes. Further details and intuition behind these constructions are provided in the technical overview below.

### 1.1.1 Private vs. Public Randomness

Following Gao et al. [2021a] and Hanneke et al. [2022], we distinguish between two natural types of adversaries, according to their access to the learner's randomness: private-randomness adversaries and public-randomness adversaries. In the *private randomness* setting, the adversary does not observe the learner's internal random bits when constructing its poisoning attack. In the *public randomness* setting, by contrast, the adversary has full knowledge of the learner's random seed before designing the poisoned dataset.

At first glance, one might expect public randomness to reduce to the deterministic case: indeed, once the random seed is fixed, the learner becomes a deterministic function of its (poisoned) input, seemingly vulnerable to the same attacks as any deterministic algorithm. Perhaps surprisingly, this intuition proves false: although the adversary sees the learner's random seed when crafting the poisoned dataset, the underlying distribution over examples is fixed in advance and independently of this randomness. This asymmetry allows, with appropriate design, the construction of learners whose guarantees under public randomness match those achievable under private randomness:

> **Theorem 2** (Private vs. Public Randomness [2] ). *For every learning rule $\mathcal{A}_{\text{priv}}$, there exists a learning rule $\mathcal{A}_{\text{pub}}$ such that for any distribution $\mathcal{D}$, sample size $n$, and poisoning budget $\eta \in (0,1)$,*
>
> $$\max_{\substack{\text{adversary sees} \\ \text{internal randomness}}} ExcessLoss(\mathcal{A}_{\text{pub}}, \mathcal{D}, n, \eta) \leq \max_{\substack{\text{adversary does not see} \\ \text{internal randomness}}} ExcessLoss(\mathcal{A}_{\text{priv}}, \mathcal{D}, n, \eta).$$

___________
[2] see Section 5.2 for formal version.

Thus, the theorem implies that whatever guarantees can be achieved against adversaries that do not observe the learner's internal randomness can also be achieved against adversaries who do. Specifically, given any learning rule $\mathcal{A}_{\mathtt{priv}}$ that achieves a certain excess error against weak adversaries (private randomness), we can efficiently construct a learning rule $\mathcal{A}_{\mathtt{pub}}$ that achieves the same guarantees against strong adversaries (public randomness). In particular, applying the theorem to an optimal learner $\mathcal{A}_{\mathtt{priv}}^{\star}$ for the private randomness model yields a learner $\mathcal{A}_{\mathtt{pub}}^{\star}$ that matches its performance even when the adversary is fully aware of the learner's random bits.

The proof of Theorem 2 relies on carefully coupling the predictions of $\mathcal{A}_{\mathtt{priv}}$ across all possible input samples. Specifically, we construct $\mathcal{A}_{\mathtt{pub}}$ to satisfy a *monotonicity property*: whenever $\mathcal{A}_{\mathtt{priv}}$ is more likely to make an error on a test point $x$ given training sample $S_1$ than given training sample $S_2$, the learner $\mathcal{A}_{\mathtt{pub}}$ will err on $x$ for $S_1$ whenever it errs on $x$ for $S_2$. This ensures that adversaries gain no additional advantage from observing the learner's internal randomness.

### 1.1.2 Poisoned Learning Curves

The lower bound in Theorem 1 holds in the standard distribution-free PAC setting: for every learner and every sample size $n$, there exists a distribution *tailored to the sample size $n$* and to the learner that forces a large excess error. This is typical for PAC-style lower bounds, where the hard distribution is allowed to depend on the training set size.

However, in many practical learning scenarios, we think of the underlying distribution as fixed and study the learner's behavior as the sample size $n$ increases — the so-called *learning curve*. This naturally raises the question: is it possible that for any fixed distribution, as $n$ grows, the excess error under instance-targeted poisoning eventually falls below $\sqrt{d}\eta$? In other words, can better asymptotic behavior be achieved in the universal learning setting [Bousquet, Hanneke, Moran, van Handel, and Yehudayoff, 2021]? The following result shows that the answer is unfortunately negative.

> **Theorem 3** (Poisoned Learning Curves ). *Let $\mathcal{H}$ be a concept class with VC dimension $d$, and let $\eta \in (0, 1)$ be the poisoning budget. Then, for every learning rule $\mathcal{A}$, there exists a distribution $\mathcal{D}$ and an adversary that forces an excess error of at least $\Omega\left(\min\left\{\sqrt{d\eta}, 1\right\}\right)$ for infinitely many sample sizes $n$.*

Theorem 3 follows by exploiting the proof of Theorem 1. In that proof, we construct a finite set of distributions with the property that, for every learner and every sample size $n$, there exists a distribution in the set witnessing the lower bound of Theorem 1 at sample size $n$. Since the set is finite, by a simple pigeonhole principle argument, one of these distributions must witness the lower bound for infinitely many values of $n$, thus establishing Theorem 3.

### Organization

The remainder of the manuscript is organized as follows. In Section 2, we provide a technical overview of our approach, focusing on the key ideas behind the proofs. Section 3 discusses related work and places our contributions in context. In Section 4, we formalize the learning model and introduce the adversarial losses studied in this work. In Section 5 we formally state and prove our main results.

## 2  Technical Overview

In this section, we provide a high-level overview of the proof of our main result, Theorem 1. Theorems 2 and 3 are not included in this overview, as their proofs are shorter and structurally simpler than that of the main result; they are deferred to the supplementary material.

## 2.1 Lower Bound

A natural starting point for understanding the lower bound is the coin problem, introduced by Hanneke et al. [2022].[4] In this setting, the learner is shown a training set consisting of $n$ coin tosses, and must predict the outcome of a particular test toss. The adversary is allowed to observe both the training set and the target toss, and can corrupt a small fraction of the training data before the learner sees it. The learner's goal is to guess the target toss correctly despite this targeted poisoning. Formally, the coin problem is equivalent to learning a hypothesis class $\mathcal{H} = h_0, h_1$ over a single point $\mathcal{X} = \{x\}$, where $h_i(x) = i$.

**The Poisoned Coin Problem**

1. The adversary selects a bias parameter $p \in [0, 1]$.

2. A training sample $S$ of $n$ examples is drawn i.i.d. from a Bernoulli distribution of bias $p$. A target label $y$ is drawn independently from the same distribution.

3. After observing both $S$ and $y$, the adversary modifies at most an $\eta$-fraction of $S$ to obtain a poisoned sample $S'$.

4. The learner receives $S'$ and outputs a prediction $\mathcal{A}(S') \in \{0, 1\}$.

5. The learner succeeds if $\mathcal{A}(S') = y$, and fails otherwise.

In the absence of poisoning, the optimal strategy for the learner is simple: if $p > \frac{1}{2}$, the learner should always predict 1, resulting in an error rate of $1 - p$; if $p < \frac{1}{2}$, the learner should predict 0, with an error rate of $p$. Thus, while the learner does not observe the bias $p$ directly, it can estimate it from the sample. In the absence of poisoning, this allows the learner to achieve an error arbitrarily close to $\min(p, 1 - p)$ as the sample size increases.

When poisoning is allowed, the learner instead observes a corrupted sample $S'$, and we measure its performance relative to the clean optimum. We define the *excess error* as

$$\texttt{excess} = \mathbb{P}(\mathcal{A}(S') \neq y) - \min(p, 1 - p).$$

Our goal is to show that against any learner, there exists an adversary that forces an excess error of $\Omega(\sqrt{\eta})$. Toward this end, we consider the function $F(p) = \mathbb{E}[\mathcal{A}(S)]$, where $S$ is a clean (unpoisoned) training sample drawn from a Bernoulli distribution with bias $p$. That is, $F(p)$ represents the expected prediction of the learner when trained on clean samples with bias $p$. It is then convenient to consider the oblivious setting, which can be summarized as follows

**The Poisoned Coin Problem with Oblivious Adversary**

1. The adversary selects a bias parameter $p \in [0, 1]$.

2. A target label $y$ is drawn from a Bernoulli distribution of bias $p$.

3. After observing $y$, the adversary modifies $p$ into $p'$ which is close $\mathtt{d}(p, p') \leq \eta$.

4. The learner receives a training sample $S'$ of $n$ i.i.d. examples drawn from a Bernoulli distribution of bias $p'$, and outputs a prediction $\mathcal{A}(S') \in \{0, 1\}$.

5. The learner succeeds if $\mathcal{A}(S') = y$, and fails otherwise.

One can change a sample $S$ drawn from a distribution with bias $p$ to a sample $S'$ which is indistinguishable from one drawn from a distribution with bias $p'$, Thus it is suffice to find lower bounds for the oblivious setup. The advantage of this model is that the excess error can be defined in terms of the function $F$, hence the lower bound can be derived purely by analyzing properties of such functions.

Fix any learner $\mathcal{A}$. We can assume that its excess error in the absence of poisoning is at most $\sqrt{\eta}$, since otherwise the lower bound is immediate. In particular, this implies that when $p = \frac{1}{2} + \sqrt{\eta}$, the

---

[4]We note that in Hanneke et al. [2022], the coin problem was considered for a deterministic learner, while here we are dealing with a nondeterministic one. Despite a seeming resemblance, these end up being very different problems with very different answers. In particular, for the deterministic learner, a non-trivial adversary can always force an excess error of $1/2$, provided the sample size is big enough (compare it with Theorem 1).

learner's expected prediction $F(p)$ must be close to 1, and when $p = \frac{1}{2} - \sqrt{\eta}$, $F(p)$ must be close to 0. Thus, as $p$ varies from $\frac{1}{2} - \sqrt{\eta}$ to $\frac{1}{2} + \sqrt{\eta}$, the function $F(p)$ must change by a constant amount (independent of $\eta$ and $n$).

By an averaging argument, this implies that over an interval of length $O(\eta)$, the function $F(p)$ must change by at least $\Omega(\sqrt{\eta})$ on average. In particular, there exists a critical point $p^\star \in [\frac{1}{2} - \sqrt{\eta}, \frac{1}{2} + \sqrt{\eta}]$ such that

$$|F(p^\star - \eta) - F(p^\star + \eta)| = \Omega(\sqrt{\eta}).$$

This critical bias $p^\star$ will correspond to a hard instance for the learner, enabling the adversary to enforce the desired lower bound on the excess error.

The preceding argument establishes the desired lower bound for classes of VC dimension 1. Recall that a class $\mathcal{H}$ has VC dimension 1 if and only if there exists a point $x$ that is shattered by $\mathcal{H}$—meaning that for every label $y \in \{0, 1\}$, there exists a hypothesis $h \in \mathcal{H}$ such that $h(x) = y$. Thus, learning $\mathcal{H}$ on distributions supported on $x$, reduces to distinguishing between two competing hypotheses on a single point, corresponding exactly to the coin problem described above.

The case of VC dimension $d$ corresponds naturally to a generalization that we call the *d-coin problem*. In the $d$-coin problem, there are $d$ distinct coins $x_1, \ldots, x_d$, each associated with its own unknown bias $p_i \in [0, 1]$. The data generation process proceeds as follows:

**The $d$-Coin Problem**

Consider the following sampling process:

- The adversary selects a bias *vector* $p_1, \ldots, p_d \in [0, 1]^d$.
- A coin $x = x_i$ is selected uniformly at random from $\{x_1, \ldots, x_d\}$.
- The label $y$ is drawn according to a Bernoulli distribution with bias $p_i$ (that is, $y = 1$ with probability $p_i$).

The rest is as before. The training set of $n$ i.i.d. examples $(x_i, y_i)$ is generated according to the sampling process, independently of the target example $(x, y)$. After observing both the training set and the target example, the adversary may modify up to an $\eta$-fraction of the training examples to produce a poisoned sample. The learner receives the poisoned sample and attempts to predict the label $y$ of the target coin $x$. The learner succeeds if its prediction matches $y$, and fails otherwise.

We further note that, similalrly to how it was in the one coin case, the lower bound is proven using $d$-coin problem *with oblivious advrsary*, defined by analogy.

This $d$-coin problem mirrors the setting of learning a class $\mathcal{H}$ with VC dimension $d$. Indeed, if $\{x_1, \ldots, x_d\}$ is a set shattered by $\mathcal{H}$, then the label of each point $x_i$ can behave independently of the others. Assigning biases $p_i$ to the labels of each $x_i$ corresponds to constructing a distribution over labeled examples consistent with $\mathcal{H}$. Thus, the problem of predicting the label of a randomly chosen target point (under potential data poisoning) mirrors the challenge faced by the learner in the $d$-coin setting.

**The Naive Extension to $d$ Coins.** At first glance, it may seem that the lower bound for a single coin should extend directly to the $d$-coin problem. Since each coin $x_i$ has its own independent bias $p_i$, one might expect that the learner's prediction for $x_i$ depends only on the outcomes of examples corresponding to $x_i$ in the training sample, and not on examples involving other coins.

Under this assumption, the adversary's strategy would be simple: conditioned on the target coin being $x_i$, the adversary would focus its poisoning efforts solely on the training examples involving $x_i$. Since the meta-distribution is uniform over the $d$ coins, the expected number of appearances of $x_i$ in the training sample is about $n/d$. The adversary's global budget allows corrupting an $\eta$-fraction of the $n$ examples, and thus roughly a $\eta d$-fraction of the $x_i$ examples. Thus, for the target coin, the setting effectively reduces to the single-coin case with sample size about $n/d$ and poisoning budget about $\eta d$. Applying the single-coin lower bound then suggests that the excess error should be at least $\Omega\left(\sqrt{d\eta}\right)$, matching the desired bound.

Unfortunately, the above reasoning overlooks an important subtlety. In the $d$-coin problem, the adversary must commit to the biases $p_1, \dots, p_d$ *before* the training sample and target example are drawn. Because of this, the learner's prediction for a target coin $x_i$ could, in principle, depend on the outcomes of other coins $x_j$ ($j \neq i$), whose biases are correlated with $p_i$ through the adversary's global choice of parameters.

In particular, if the biases across different coins are not carefully chosen, the learner may be able to infer information about the bias of $x_i$ by examining patterns across the entire sample, not just the examples involving $x_i$. This possibility of *information leakage* — where the behavior of non-target coins reveals something about the target coin — breaks the reduction to the single-coin case. Thus, a more careful argument is needed to establish the desired lower bound.

**The Fix: Randomizing the Biases.** To overcome this difficulty, we modify the adversary's strategy: rather than fixing the biases $p_1, \dots, p_d$ deterministically, the adversary draws each bias independently at random from a carefully designed distribution over $[0, 1]$. This randomization breaks potential correlations between different coins, ensuring that the behavior of non-target coins carries no useful information about the target coin.

Constructing such a hard distribution requires strengthening the lower bound for the single-coin case ($d = 1$): instead of selecting a hard bias $p$ tailored to a specific learner, we design a *distribution over biases* that is universally hard — meaning that for any learner, the expected excess error (over the choice of the bias) remains $\Omega(\sqrt{\eta})$. Sampling the biases $p_1, \dots, p_d$ independently from this hard distribution ensures that, for any fixed learner, the expected excess error remains large, and prevents information leakage between coins. With this setup, the adversary effectively reduces the $d$-coin problem back to $d$ independent copies of the single-coin case, yielding the desired $\Omega(\sqrt{d\eta})$ lower bound.

## 2.2 Upper Bound

**Coin Problem.** To build intuition for the upper bound, we begin with the coin problem. One natural way to exploit randomness in the learner's strategy is via sub-sampling. Specifically, the learner can randomly select a small sub-sample of size $k \sim \frac{1}{\eta}$ from the training set and predict the label of the test coin by majority vote over this sub-sample.

This simple strategy has two key advantages. First, by standard concentration bounds, a small sub-sample already suffices to estimate the bias of the coin up to a small additive error. Second, and crucially, by anti-concentration, the adversary cannot easily flip the prediction: changing the majority outcome typically requires modifying roughly $\sqrt{k}$ entries. As long as the poisoned fraction $\eta$ is small, the adversary is unlikely to control enough points in the sub-sample to alter the majority. In total, this approach yields an excess error of $O(\sqrt{\eta})$ in the coin problem.

**Finite Classes.** A natural next step is to generalize this idea to arbitrary finite hypothesis classes. One might try the same strategy: draw a small random sub-sample of size $k$ and train an optimal PAC learner on it. This technique was used by Gao et al. [2021a], who showed it suffices for learning under instance-targeted poisoning when the poisoned fraction $\eta$ vanishes with $n$. However, this method fails to provide our desired bounds when $\eta$ is not negligible.

What fails here is robustness. In the coin problem, majority vote has a useful anti-concentration property: to flip the output, the adversary must corrupt roughly $\sqrt{k}$ points. But for general hypothesis classes, it is unclear whether any natural learning rule exhibits similar resilience to perturbations. In particular, standard PAC learners might change their output significantly in response to a few targeted changes, especially when trained on a small sub-sample. This instability limits the effectiveness of naive sub-sampling in the general case.

**Sampling via Loss Exponentiation.** To move beyond naive sub-sampling, it is helpful to ask: what probability distribution over hypotheses does the sub-sampling strategy induce?

In the coin problem, majority voting over a random sub-sample can be interpreted as assigning higher selection probability to the constant label $h \equiv 0$ or $h \equiv 1$) that achieves lower empirical loss. More quantitatively, one can show that the induced selection probability is roughly proportional to

the exponential of the negative squared loss on the sub-sample: $\mathbb{P}(h) \propto \exp\left(-\lambda \cdot \hat{L}_S(h)^2\right)$, for some $\lambda > 0$. This reflects the anti-concentration property of the majority vote: flipping the output requires altering many points in the sub-sample (although we note that due to the abovementioned problems, we end up not using the anti-concentration inequalitis per se).

This motivates the use of learners that explicitly reweight hypotheses according to exponentiated losses. In our final algorithm for finite classes, we simplify the squared loss to standard loss and sample from the exponential of its negative: $\mathbb{P}(h) \propto \exp\left(-\lambda \cdot \hat{L}_S(h)\right)$. This distribution, known from the exponential mechanism in differential privacy and multiplicative weights in online learning, preserves both stability and performance. Small perturbations to the dataset (such as an $\eta$-fraction of poisonings) have limited impact on the output, while hypotheses with lower empirical error remain more likely to be selected. Working out the details of this approach yields an excess error bound of $\widetilde{O}(\sqrt{\eta \log m})$ for finite hypothesis classes of size $m$.

**From Finite to VC Classes.** All that remains is to extend the result from finite classes to hypothesis classes of bounded VC dimension. In the case of a finite class $\mathcal{H}$ of size $m$, the exponential sampling strategy achieves excess error $\tilde{O}(\sqrt{\eta \log m})$. Our goal is to replace the dependence on $\log m$ with $\mathrm{VC}(\mathcal{H})$, the VC dimension of the class.

A direct application of the multiplicative weights sampling method to an infinite class fails to yield tight bounds, as the performance of the learner would then scale with the (possibly infinite) size of the class, rather than with its VC dimension.

To overcome this, we use a classical reduction based on $\varepsilon$-covers: we construct a finite cover $\mathcal{H}' \subseteq \mathcal{H}$ such that the minimal loss over $\mathcal{H}'$ approximates that over $\mathcal{H}$ up to an additive $\varepsilon$. This allows us to reduce to the finite case without significantly increasing the error.

A standard uniform convergence argument shows that an $\varepsilon$-cover can be constructed from a random sample of size roughly $\mathrm{VC}(\mathcal{H})/\varepsilon^2$. Unfortunately, using such a sample leads to a suboptimal overall bound on the excess error. Instead, we apply a more refined analysis based on the VC dimension of the symmetric difference class $h \triangle h' : h, h' \in \mathcal{H}$, which shows that it suffices to construct the $\varepsilon$-cover from a much smaller subsample of size $\tilde{O}(\mathrm{VC}(\mathcal{H})/\varepsilon)$. This enables us to keep the size of the subsample below the poisoning threshold with high probability, ensuring that the cover is not corrupted.

A similar approach — involving subsampling and refined covering arguments — has been used in the context of differential privacy and algorithmic stability, for example in Bassily et al. [2019], Dagan and Feldman [2020].

Putting everything together, our final algorithm proceeds as follows:

1. Given a sample $S$, draw a random sub-sample $T$ of size $\tilde{O}(\sqrt{\mathrm{VC}(\mathcal{H})/\eta})$.

2. Use $T$ to construct a finite $\varepsilon$-cover $\mathcal{H}_T \subseteq \mathcal{H}$, where $\varepsilon = \sqrt{\mathrm{VC}(\mathcal{H}) \cdot \eta}$.

3. Run exponential sampling over $\mathcal{H}_T$: select $h \in \mathcal{H}_T$ with probability proportional to $\exp(-\lambda \cdot \hat{L}_S(h))$.

This completes the proof sketch of the upper bound.

# 3   Related Work

Learning under poisoning attacks has been studied in several settings. Earlier research [Valiant, 1985, Kearns and Li, 1993, Sloan, 1995, Bshouty, Eiron, and Kushilevitz, 2002] focused on *non-targeted* poisoning, where the adversary does not know the test point. Computational aspects of efficient learning under poisoning have been studied under various distributional and algorithmic assumptions [Kalai, Klivans, Mansour, and Servedio, 2008, Klivans, Long, and Servedio, 2009, Awasthi, Balcan, and Long, 2014, Diakonikolas, Kane, and Stewart, 2018]. In particular, Awasthi, Balcan, and Long [2014] achieved nearly optimal learning guarantees (up to constant factors) for polynomial-time algorithms learning homogeneous linear separators under distributional assumptions in the malicious noise model. These results were later extended to the *nasty noise* model by Diakonikolas, Kane, and Stewart [2018], along with techniques that also apply to other geometric

concept classes. In the *unsupervised* setting, the computational challenges of learning under poisoning attacks have been investigated by Diakonikolas, Kamath, Kane, Li, Moitra, and Stewart [2016], Lai, Rao, and Vempala [2016].

In contrast, our work focuses on *instance-targeted* poisoning, and investigates the fundamental tradeoffs in error and sample complexity, independent of computational constraints. A related but more demanding task—certifying the *correctness* of individual predictions under instance-targeted poisoning—was studied by Balcan, Blum, Hanneke, and Sharma [2022]. The instance-targeted model we study was formalized in Gao, Karbasi, and Mahmoody [2021b], who showed that when the poisoning budget $\eta$ vanishes with the sample size, PAC learnability is preserved. Hanneke, Karbasi, Mahmoody, Mehalel, and Moran [2022] extended this line of work to general (non-vanishing) poisoning rates, and characterized the optimal error in the realizable setting. In particular, they showed that deterministic learners suffice in the realizable case, but fail in the agnostic case, where they can suffer near-maximal error even under minimal poisoning. They left open the question of whether *randomized* learners could succeed in the agnostic case. Our work resolves this open problem, establishing that randomized learners can indeed achieve meaningful guarantees in the agnostic setting, and characterizing the optimal excess error as $\widetilde{\Theta}(\sqrt{d\eta})$, where $d$ is the VC dimension.

Other types of targeted attacks have also been studied. In *model-targeted* attacks, the adversary aims to force the learner to mimic a particular model [Farhadkhani, Guerraoui, Hoang, and Villemaud, 2022, Suya, Mahloujifar, Suri, Evans, and Tian, 2021]. *Label-targeted* attacks aim to flip the learner's prediction on a specific test example [Chakraborty, Alam, Dey, Chattopadhyay, and Mukhopadhyay, 2018]. Jagielski, Severi, Pousette Harger, and Oprea [2021] introduced *subpopulation* poisoning, where the adversary knows that the test point comes from a specific subset of the population.

More broadly, recent works have explored robustness to adversarial test-time manipulation. Balcan, Hanneke, Pukdee, and Sharma [2023] study reliable prediction under adversarial test-time attacks and distribution shifts. While their work focuses on modifying the test point rather than poisoning the training set, it shares our motivation of provable robustness in challenging environments. Like-wise, Goel, Hanneke, Moran, and Shetty [2023] analyze a sequential learning model with clean-label adversaries, allowing abstention on uncertain inputs.

Empirical and algorithmic defenses against instance-targeted and clean-label poisoning have been widely studied. Rosenfeld, Winston, Ravikumar, and Kolter [2020] show that randomized smoothing [Cohen, Rosenfeld, and Kolter, 2019] can mitigate label-flipping attacks, and can extend to replacing attacks such as ours. Follow-up work has explored deterministic defenses [Levine and Feizi, 2020], as well as randomized sub-sampling and bagging [Chen, Li, Wu, Sheng, and Li, 2020, Weber, Xu, Karlas, Zhang, and Li, 2020, Jia, Cao, and Gong, 2020].

Other theoretical works have studied error amplification under targeted poisoning [Mahloujifar and Mahmoody, 2017, Etesami, Mahloujifar, and Mahmoody, 2020], often with a focus on specific test examples. The empirical study of poisoning attacks in Shafahi, Huang, Najibi, Suciu, Studer, Dumitras, and Goldstein [2018] further illustrates the practical relevance of instance-targeted threats.

## 4   Preliminaries

As it is usual in learning theory, we consider a concept class $\mathcal{H}$ over domain $\mathcal{X}$ with a label space $\mathcal{Y} = \{0, 1\}$; that is, $\mathcal{H}$ is a set of functions (also called concepts) from $\mathcal{X}$ to $\mathcal{Y}$. We define the set of labeled examples as $\mathcal{Z} = \mathcal{X} \times \mathcal{Y}$ and a *sample of size* $n$ as a sequence $S \in \mathcal{Z}^n$; the space of samples is $\mathcal{Z}^\star = \bigcup_{n=1}^\infty \mathcal{Z}^n$. A learning rule is a map $\mathcal{A} : \mathcal{Z}^\star \to \mathcal{Y}^{\mathcal{X}}$ that assigns to each sample $S$ a function $\mathcal{A}(S)$, called a *hypothesis*. We also often consider randomized learners that output a distribution over hypotheses.

The *sample loss* of a hypothesis $h : \mathcal{X} \to \mathcal{Y}$ on a sample $S \in \mathcal{Z}^n$ is

$$L_S(h) := \frac{1}{n} \sum_{i=1}^n |h(x_i) - y_i|.$$

Similarly, we define the *population loss* of $h$ with respect to a distribution $\mathcal{D}$ over $\mathcal{Z}$ as

$$L_{\mathcal{D}}(h) = \mathbb{P}_{(x,y)\sim\mathcal{D}}[h(x) \neq y] = \mathbb{E}_{(x,y)\sim\mathcal{D}} |h(x) - y|.$$

The expected loss of a learner $\mathcal{A}$ with a sample size $n$ with respect to $\mathcal{D}$ is

$$L_{\mathcal{D}}(\mathcal{A}, n) = \mathop{\mathbb{E}}_{S \sim \mathcal{D}^n} L_{\mathcal{D}}\big(\mathcal{A}(S)\big) = \mathop{\mathbb{E}}_{\substack{S \sim \mathcal{D}^n \\ (x,y) \sim \mathcal{D}}} |\mathcal{A}(S)(x) - y|.$$

The above formula is also applicable to randomized learners, in which case the expectation is also over the internal randomness of $\mathcal{A}$.

Define the normalized Hamming distance between two samples $S, S' \in \mathcal{Z}^n$ by

$$\mathtt{d}_H(S, S') = \frac{1}{n} |\{i \in [n] \; : \; S_i \neq S'_i\}| \,.$$

For any sample $S \in \mathcal{Z}^n$ and $\eta \in (0, 1)$, define the $\eta$-ball centered at $S$ by $B_\eta(S) = \{S' \in \mathcal{Z}^n \; : \; \mathtt{d}_H(S, S') \leq \eta\}$.

**Definition 1** ($\eta$-adversarial loss). *Let $\eta \in (0, 1)$ be the adversary's budget, let $\mathcal{A}$ be a (possibly randomized) learning rule, and let $\mathcal{D}$ be a distribution over examples. The $\eta$-adversarial loss of $\mathcal{A}$ with sample size $n$ with respect to $\mathcal{D}$ is defined as*

$$L_{\mathcal{D},\eta}(\mathcal{A}, n) = \mathop{\mathbb{E}}_{\substack{S \sim \mathcal{D}^n \\ (x,y) \sim \mathcal{D}}} \left[ \sup_{S' \in B_\eta(S)} \mathop{\mathbb{E}}_r |\mathcal{A}_r(S')(x) - y| \right],$$

*where $r$ is the internal randomness of $\mathcal{A}$, and $\mathbb{E}_r$ can be omitted in case the learner $\mathcal{A}$ is deterministic.*

Note that in this definition the data is corrupted before the randomness of $\mathcal{A}$, which corresponds to private randomness. Alternatively, we may consider the public randomness model in which we define

$$L_{\mathcal{D},\eta}^{\mathtt{pub}}(\mathcal{A}, n) = \mathop{\mathbb{E}}_{\substack{S \sim \mathcal{D}^n \\ (x,y) \sim \mathcal{D}}} \mathop{\mathbb{E}}_r \left[ \sup_{S' \in B_\eta(S)} |\mathcal{A}_r(S')(x) - y| \right].$$

Finally, in the spirit of agnostic learning and the poisoned coin problem, the effectiveness of the learner in the presence of poisoning is measured with respect to excess error:

**Definition 2** (Excess error). *For a concept class $\mathcal{H}$, adversary budget $\eta \in (0, 1)$, learning rule $\mathcal{A}$, sample size $n$, and a distribution over examples $\mathcal{D}$, we define the* excess error *of $\mathcal{A}$ with sample size $n$ against an adversary with budget $\eta$ on $\mathcal{D}$ as*

$$\mathtt{excess}_{\mathcal{H},\mathcal{D},\eta}(\mathcal{A}, n) = L_{\mathcal{D},\eta}(\mathcal{A}, n) - \inf_{h \in \mathcal{H}} L_{\mathcal{D}}(h).$$

The definition of excess is similarly adapted to the case of public randomness, and, as usual, we drop some of the parameters $\mathcal{H}, \mathcal{D}, \eta, \mathcal{A}$, or $n$ in the notation for loss/excess if they are clear from the context.

**Definition 3** (VC dimension). *We say that a concept class $\mathcal{H}$ shatters a set $X \subset \mathcal{X}$ if for any function $f : X \to \{0, 1\}$ there exists $h \in \mathcal{H}$ such that $f(x) = h(x)$ for all $x \in X$. The VC dimension of $\mathcal{H}$, denoted $\mathtt{VC}(\mathcal{H})$, is the largest $d$ for which there exist a set $X \subset \mathcal{X}$ of size $d$ that is shattered by $\mathcal{H}$. If sets of arbitrary size can be shattered we define $\mathtt{VC}(H) = \infty$.*

**Lemma 4** (Sauer–Shelah lemma - see Lemma 6.10 in Shalev-Shwartz and Ben-David [2014]). *Let $\mathcal{H}$ be a concept class of VC dimension $d = \mathtt{VC}(\mathcal{H})$, and let $X \subset \mathcal{X}$ be a finite set of unlabeled examples of size $n = |X|$. Define an equivalence relation on $\mathcal{H}$ by $h \sim_X h'$ if $h(x) = h'(x)$ for all $x \in X$, and let $\mathcal{H}_X$ be an arbitrary set of representatives with respect to this relation. Then we have*

$$|\mathcal{H}_X| \leq \sum_{i=0}^{d} \binom{n}{d}.$$

*In particular, if $n > d + 1$ we have*

$$|\mathcal{H}_X| \leq (\frac{e \cdot n}{d})^d.$$

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

# 5 Proofs

## 5.1 Proof of Theorem 1

We prove Theorem 1 in a form of Theorem 4 below, which gives quantitative version of the bounds announced in it.

**Theorem 4** (Main result - quantitative version). *Let $\mathcal{H}$ be a concept class with VC dimension $d \geq 1$ and suppose $\eta < \frac{1}{4d}$. Then there exits a randomized learner $\mathcal{A}$ such that for any distribution $\mathcal{D}$ on $\mathcal{X} \times \mathcal{Y}$ and all $n \geq \frac{1}{\eta}$ we have:*

$$\mathrm{excess}_{\mathcal{H},\mathcal{D},\eta}(\mathcal{A},n) \leq 36\sqrt{\eta d}\log\frac{e}{\eta d}.$$

*This bound is also tight up to log factors; that is, for every randomized learner $\mathcal{A}$, $\eta < \frac{1}{d}$, and $n \geq \frac{6}{\eta}\log\frac{64}{\sqrt{d\eta}}$, there exists a distribution $\mathcal{D}$ such that*

$$\mathrm{excess}_{\mathcal{H},\mathcal{D},\eta}(\mathcal{A},n) \geq \frac{\sqrt{d\eta}}{36}.$$

There will be no separate proof of Theorem 4. Instead, the entire Section 5.1 is treated as such proof. In particular, the upper bound on the excess is established in Lemma 9, and the lower bound in Lemma 14. We also remark that Theorem 1 is a direct consequence of Theorem 4. Indeed the bounds in Theorem 4 are quantitative versions of the ones in Theorem 1 so we just need to justifies the assumptions of Theorem 4.

For the upper bound the assumption $\eta < \frac{1}{4d}$ is safe since otherwise a bound of $\tilde{O}(\sqrt{d\eta}) = O(1)$ is trivial, while the assumption $\eta < \frac{1}{n}$ was already assumed in Theorem 1, where we remarked that without it the adversary is unable to poison even a single point.

For the lower bound, we may assume $\eta < \frac{1}{d}$, since otherwise $\sqrt{d\eta} = \Omega(1)$ and the result is clear. The assumption $n \geq \frac{6}{\eta}\log\frac{64}{\sqrt{d\eta}}$ is valid since without it, standard PAC learning arguments shows that the desired lower bounds can be forced even without poisoning. Indeed, by 28.2.22 in Shalev-Shwartz and Ben-David [2014], an excess loss of $\Omega(\sqrt{\frac{d}{n}})$, can always be forced, even without poisoning. In the case of $n \leq \frac{6}{\eta}\log\frac{64}{\sqrt{d\eta}}$, this becomes the desired $\tilde{\Omega}(\sqrt{d\eta})$.

### 5.1.1 Upper bound

We address loss of learners with poisoning via *prediction stable* learners:

**Definition 5** (Prediction stability). *The $\eta$-prediction stability of a random learner $\mathcal{A}$ with sample size $n$ with respect to a distribution $\mathcal{D}$ is*

$$\lambda_n(\mathcal{A}|\mathcal{D},\eta) = \mathop{\mathbb{E}}_{\substack{S\sim\mathcal{D}^n \\ (x,y)\sim\mathcal{D}}}\left[\sup_{S'\in B_\eta(S)}\mathop{\mathbb{P}}_r\left[\mathcal{A}_r(S)(x) \neq \mathcal{A}_r(S')(x)\right]\right]$$

$$= \mathop{\mathbb{E}}_{\substack{S\sim\mathcal{D}^n \\ (x,y)\sim\mathcal{D}}}\left[\sup_{S'\in B_\eta(S)}\mathop{\mathbb{E}}_r\left|\mathcal{A}_r(S')(x) - \mathcal{A}_r(S)(x)\right|\right].$$

Note that, trivially,

$$\max\left(\lambda_n(\mathcal{A}|\mathcal{D},\eta), L_\mathcal{D}(\mathcal{A},n)\right) \leq L_{\mathcal{D},\eta}(\mathcal{A},n) \leq \lambda_n(\mathcal{A}|\mathcal{D},\eta) + L_\mathcal{D}(\mathcal{A},n),$$

where the second inequality can be trivially rewritten as

$$\mathrm{excess}_{\mathcal{H},\mathcal{D},\eta}(\mathcal{A},n) \leq \lambda_n(\mathcal{A}|\mathcal{D},\eta) + \mathrm{excess}_{\mathcal{H},\mathcal{D}}(\mathcal{A},n), \tag{1}$$

where $\mathrm{excess}_{\mathcal{H},\mathcal{D}}(\mathcal{A},n) = L_\mathcal{D}(\mathcal{A}_\mathcal{H},n) - \inf_{h\in\mathcal{H}} L_\mathcal{D}(h)$.

So minimizing the loss of a learner with poisoning is equivalent to finding an efficient learner that is prediction stable.

**Proposition 6.** *Let $\mathcal{H} \subseteq \mathcal{Y}^{\mathcal{X}}$ be a finite concept class of size $|\mathcal{H}| = m$ and let $\eta > 0$ be the poisoning budget. Then there exists a randomized learner $\mathcal{A}_{\mathcal{H}}$ such that for every distribution $\mathcal{D}$ over $\mathcal{Z}$ and every $n > 0$, it holds*

$$\lambda_n(\mathcal{A}_{\mathcal{H}}|\mathcal{D},\eta) \leq 4\sqrt{\eta \log m},$$

$$\mathtt{excess}_{\mathcal{H},\mathcal{D}}(\mathcal{A}_{\mathcal{H}}, n) \leq \sqrt{\eta \log m} + 4\sqrt{\frac{\log 2m}{n}}.$$

*Proof.* We define the learner $\mathcal{A}_{\mathcal{H}}$ in two stages. First, we define the learner $\mathcal{A}$, that samples according to the exponential mechanism with multiplicative weights. That is, it outputs each $h \in \mathcal{H}$ with probability

$$\mathbb{P}\left(\mathcal{A}(S) = h\right) = \frac{e^{-tL_S(h)}}{W},$$

where $W = W(S) = \sum_{h \in \mathcal{H}} e^{-tL_S(h)}$ and $t = \sqrt{\frac{\log m}{\eta}}$. We then prove that $\mathcal{A}$ satisfies the bound on the loss from the statement of the proposition, that is, that $L_{\mathcal{D}}(\mathcal{A}, n) \leq \inf_{h \in \mathcal{H}} L_{\mathcal{D}}(h) + 5\sqrt{\eta \log 2m}$.

As our second stage, we define another learner $\mathcal{B}$, which will be our target learner, as follows: $\mathcal{B}$ samples $r \sim \mathcal{U}(0,1)$. Then, for a sample $S$ and $x \in \mathcal{X}$, $\mathcal{B}$ returns 1 if $r \leq \mathbb{P}_r\left[\mathcal{A}_r(S)(x) = 1\right]$, and 0 otherwise. It is easy to see that $\mathcal{B}$ is a different learner than $\mathcal{A}$; in particular, $\mathcal{B}$ can output hypotheses outside of the class $\mathcal{H}$. However, by construction, for all $S$ and $x$, it holds

$$\mathbb{P}_r\left[\mathcal{B}_r(S)(x) = 1\right] = \mathbb{P}_r\left[\mathcal{A}_r(S)(x) = 1\right].$$

In particular, this easily implies that $L_{\mathcal{D}}(\mathcal{B}, n) = L_{\mathcal{D}}(\mathcal{A}, n)$, and so $\mathcal{B}$ also satisfies the required bound on the loss. Notice, however, that, unlike with $\mathcal{A}$, for $\mathcal{B}$ there is an explicit dependence of the output on the internal randomness $r$. Thus, the outputs of $\mathcal{B}$ on any samples $S$ and $S'$ are naturally coupled. We will then utilize this coupling to prove the required bound on the prediction stability of $\mathcal{B}$.

Let us now start with the proof of the bound on $L_{\mathcal{D}}(\mathcal{A}, r)$. The proof proceeds by several claims.

*Claim 1.* For any sample $S$, it holds:

$$\mathbb{E}_r L_S\left(\mathcal{A}_r(S)\right) \leq \inf_{h \in \mathcal{H}} L_S(h) + \frac{\log m}{t}.$$

Indeed,

$$\exp\left(t \cdot \mathbb{E}_r L_S\left(\mathcal{A}_r(S)\right)\right) \leq [\text{by Jensen's}] \leq \mathbb{E}_r \exp\left(t \cdot L_S\left(\mathcal{A}_r(S)\right)\right)$$

$$= \frac{1}{W}\sum_h e^{tL_S(h)}e^{-tL_S(h)} = \frac{m}{W}$$

$$\leq \left[W = \sum_{h \in \mathcal{H}} e^{-tL_S(h)} \geq \sup_h e^{-tL_S(h)}\right] \leq m \cdot \left(\sup_{h \in \mathcal{H}} e^{-tL_S(h)}\right)^{-1} \leq m \exp\left(t \inf_{h \in \mathcal{H}} L_S(h)\right).$$

Taking the log of both sides we get

$$\mathbb{E}_r L_S\left(\mathcal{A}_r(S)\right) \leq \inf_{h \in \mathcal{H}} L_S(h) + \frac{\log m}{t},$$

as needed.

*Claim 2.*

$$\mathbb{E}_{S \sim \mathcal{D}^n} \sup_{h \in \mathcal{H}} |L_{\mathcal{D}}(h) - L_S(h)| \leq 2\sqrt{\frac{\log 2m}{n}}.$$

Indeed, for any $h \in \mathcal{H}$ and $\varepsilon > 0$ by Hoeffding's inequality we have

$$\mathbb{P}_{S \sim \mathcal{D}^n}\left(|L_S(h) - L_{\mathcal{D}}(h)| > \varepsilon\right) \leq 2e^{-\varepsilon^2 n}.$$

And, by union bound, this gives

$$\mathop{\mathbb{P}}_{S\sim\mathcal{D}^n}\left(\sup_{h\in\mathcal{H}}|L_S(h)-L_{\mathcal{D}}(h)|>\varepsilon\right)\leq 2me^{-\varepsilon^2 n}.$$

Let us define $\varepsilon_0=\sqrt{\frac{\log 2m}{n}}$, the value for which $2me^{-\varepsilon_0^2 n}=1$, and compute

$$\mathop{\mathbb{E}}_{S\sim\mathcal{D}^n}\sup_{h\in\mathcal{H}}|L_D(h)-L_S(h)|=\int_0^1\mathop{\mathbb{P}}_{S\sim\mathcal{D}^n}\left(\sup_{h\in\mathcal{H}}|L_S(h)-L_{\mathcal{D}}(h)|>t\right)dt$$

$$\leq\int_0^{\varepsilon_0}1+\int_{\varepsilon_0}^1 2me^{-t^2 n}dt\leq\left[x:=\sqrt{n}\cdot t\right]\leq\varepsilon_0+\frac{2m}{\sqrt{n}}\int_{\varepsilon_0\sqrt{n}}^{\infty}e^{-x^2}dx$$

$$\leq\left[\varepsilon_0\sqrt{n}=\sqrt{\log 2m}\geq 1,\text{ so we estimate }e^{-x^2}\leq xe^{-x^2}\right]$$

$$\leq\varepsilon_0+\frac{2m}{\sqrt{n}}\int_{\varepsilon_0\sqrt{n}}^{\infty}xe^{-x^2}dx=\varepsilon_0+\frac{2m}{\sqrt{n}}\cdot\left(-\frac{1}{2}e^{-x^2}\Big|_{\varepsilon_0\sqrt{n}}^{\infty}\right)$$

$$\leq\varepsilon_0+\frac{2m}{\sqrt{n}}\cdot\frac{\exp(-\varepsilon_0^2 n)}{2}=\sqrt{\frac{\log 2m}{n}}+\frac{m\exp(-\log 2m)}{\sqrt{n}}\leq 2\sqrt{\frac{\log 2m}{n}}.$$

This finishes the proof of Claim 2.

Putting Claims 1 and 2 together and recalling that $t=\sqrt{\frac{\log m}{\eta}}$, we get the required bound on $L_{\mathcal{D}}(\mathcal{A},n)$:

$$L_{\mathcal{D}}(\mathcal{A},n)=\mathop{\mathbb{E}}_{S\sim\mathcal{D}^n}L_{\mathcal{D}}\big(\mathcal{A}(S)\big)\leq\mathop{\mathbb{E}}_{S\sim\mathcal{D}^n}L_S\big(\mathcal{A}_{\mathcal{H}}(S)\big)+2\sqrt{\frac{\log 2m}{n}}$$

$$\leq\inf_{h\in\mathcal{H}}L_S(h)+\frac{\log m}{t}+2\sqrt{\frac{\log 2m}{n}}\leq\inf_{h\in\mathcal{H}}L_{\mathcal{D}}(h)+\sqrt{\eta\log m}+4\sqrt{\frac{\log 2m}{n}}.$$

Let us now prove the prediction stability bound for $\mathcal{B}$. Recall that $\mathcal{B}$ returns 1 if $r\leq\mathbb{P}_r\left[\mathcal{A}_r(S)(x)=1\right]$, and 0 otherwise, for $r\sim\mathcal{U}(0,1)$. Again, we start with an intermediate claim, which, somewhat unexpectedly, is still be about $\mathcal{A}$, not $\mathcal{B}$.

*Claim 3.* Let $S,S'\in\mathcal{Z}^n$ be two samples with $\mathtt{d}_H(S,S')\leq\eta$. Then, for any $h\in\mathcal{H}$, we have

$$e^{-2t\eta}\cdot\mathop{\mathbb{P}}_r[\mathcal{A}_r(S')=h]\leq\mathop{\mathbb{P}}_r[\mathcal{A}_r(S)=h]\leq e^{2t\eta}\cdot\mathop{\mathbb{P}}_r[\mathcal{A}_r(S')=h].$$

Indeed, by the definition of $\mathtt{d}_H$, $L_S(h)\leq L_{S'}(h)+\eta$, so

$$e^{-t\eta}\cdot e^{-tL_{S'}(h)}\leq e^{-tL_S(h)}\leq e^{t\eta}\cdot e^{-tL_{S'}(h)},$$

where the second inequality is by symmetry. So,

$$W=\sum_{h\in\mathcal{H}}e^{-tL_S(h)}\leq e^{t\eta}\cdot\sum_{h\in\mathcal{H}}e^{-tL_{S'}(h)}=e^{t\eta}\cdot W',$$

and so

$$\mathop{\mathbb{P}}_r[\mathcal{A}_r(S')=h]=\frac{e^{-tL_{S'}(h)}}{W'}\leq e^{2t\eta}\cdot\frac{e^{-tL_S(h)}}{W}=e^{2t\eta}\cdot\mathop{\mathbb{P}}_r[\mathcal{A}_r(S')=h].$$

And the second inequality is by symmetry. This finishes the proof of Claim 3.

Now, let $S$ and $S'$ be $\eta$-close samples and let $x \in \mathcal{X}$. Then

$$\mathbb{P}_r\big[\mathcal{B}_r(S)(x) \neq \mathcal{B}_r(S')(x)\big]$$

$$= \mathbb{P}_r\big[\mathcal{B}_r(S)(x) = 1 \text{ and } \mathcal{B}_r(S')(x) = 0\big] + \mathbb{P}_r\big[\mathcal{B}_r(S)(x) = 0 \text{ and } \mathcal{B}_r(S')(x) = 1\big]$$

$$= \left| \mathbb{P}_r\big[\mathcal{A}_r(S)(x) = 1\big] - \mathbb{P}_r\big[\mathcal{A}_r(S')(x) = 1\big] \right|$$

$$= \left| \sum_{h:\, h(x)=1} \mathbb{P}_r\big[\mathcal{A}_r(S)(x) = h\big] - \sum_{h:\, h(x)=1} \mathbb{P}_r\big[\mathcal{A}_r(S')(x) = h\big] \right|$$

$$\leq \sum_h \left| \mathbb{P}_r\big[\mathcal{A}_r(S)(x) = h\big] - \mathbb{P}_r\big[\mathcal{A}_r(S')(x) = h\big] \right|$$

$$\leq [\text{By Claim 3}] \leq \sum_h \max\Big( \mathbb{P}_r\big[\mathcal{A}_r(S)(x) = h\big], \mathbb{P}_r\big[\mathcal{A}_r(S')(x) = h\big] \Big) \cdot (1 - e^{-2t\eta})$$

$$\leq 2(1 - e^{-2t\eta}) \leq 4t\eta = 4\sqrt{\eta \log m},$$

where in the last inequality we used the estimate $1 - e^{-x} \leq x$ for $x \in (0, 1)$. The last bound clearly extends to $\lambda_n(\mathcal{B}|\mathcal{D}, \eta) \leq 4\sqrt{\eta \log m}$, as needed. Because, as argued in the beginning, $\mathcal{B}$ satisfies the same guarantee on $L_\mathcal{D}$, it thus satisfies both bounds in the statement of the proposition, finishing the proof. $\qquad\square$

**Definition 7** ($\varepsilon$-cover). *Let $\mathcal{H}$ be a concept class over domain $\mathcal{X}$, let $\varepsilon > 0$ and $\mathcal{D}$ be a distribution over $\mathcal{X}$. Then $H \subseteq \mathcal{H}$ is called an $\varepsilon$-cover of $\mathcal{H}$ with respect to $\mathcal{D}$ if for any $h \in \mathcal{H}$ there is $h' \in H$ such that $\mathbb{P}_{x \sim \mathcal{D}}\big[h(x) \neq h'(x)\big] \leq \varepsilon$. Let $\mathcal{E}_\mathcal{D}(H)$ be the minimal $\varepsilon > 0$ for which $H$ is an $\varepsilon$-cover of $\mathcal{H}$, that is*

$$\mathcal{E}_\mathcal{D}(H) = \sup_{h \in \mathcal{H}} \inf_{h' \in H} \mathbb{P}_{x \sim \mathcal{D}}\big[h(x) \neq h'(x)\big].$$

We also use the above definition with distributions $\mathcal{D}$ over examples $\mathcal{X} \times \mathcal{Y}$, rather than just over $\mathcal{X}$. This is done in a natural way, by letting $\mathcal{E}_\mathcal{D}(H) = \mathcal{E}_{\mathcal{D}_\mathcal{X}}(H)$, where $\mathcal{D}_\mathcal{X}$ is the marginal of $\mathcal{D}$.

For a set of unlabeled examples $X \subseteq \mathcal{X}$ recall we defined an equivalence relation on $\mathcal{H}$ by $h \sim_X h'$ if $h(x) = h'(x)$ for all $x \in X$, and denoted $\mathcal{H}_X$ an arbitrary set of representatives with respect to this relation. Extend this definition to labeled samples $S = \big((x_1, y_1), (x_2, y_2), \ldots, (x_n, y_n)\big)$ by letting $\mathcal{H}_S = \mathcal{H}_X$ for $X = \{x_1, \ldots, x_n\}$.

The following lemma is based on Lemma 3.3 in Bassily et al. [2019].

**Lemma 8** (Covers for VC classes). *For any concept class $\mathcal{H}$ with VC dimension $d$, for any distribution $\mathcal{D}$ and any $n \geq d$, we have:*

$$\mathbb{E}_{S \sim \mathcal{D}^n} \mathcal{E}(\mathcal{H}_S) \leq \frac{13d}{n} \log \frac{2en}{d}.$$

*Proof.* The proof relies on Lemma 3.3 in Bassily et al. [2019], specifically, we use claim (1) in this result which gives

$$\mathbb{P}_{S \sim \mathcal{D}^n}[\mathcal{E}(\mathcal{H}_S) > \varepsilon] \leq 2\big(\frac{2en}{d}\big)^{2d} \exp\big(\frac{-\varepsilon n}{4}\big).$$

Hence for any $\varepsilon > 0$ we have

$$\mathbb{E}_{S \sim \mathcal{D}^n}[\mathcal{E}(\mathcal{H}_S)] \leq \varepsilon + 2\big(\frac{2en}{d}\big)^{2d} \exp\big(\frac{-\varepsilon n}{4}\big).$$

Taking $\varepsilon = \frac{12d}{n} \log \frac{2en}{d}$ we get

$$\mathbb{E}_{S \sim \mathcal{D}^n}[\mathcal{E}(\mathcal{H}_S)] \leq \varepsilon + 2\big(\frac{2en}{d}\big)^{2d} \exp\big(-3d \log \frac{2en}{d}\big) = \frac{12d}{n} \log \frac{2en}{d} + 2\big(\frac{2en}{d}\big)^{-d} \leq \frac{13d}{n} \log \frac{2en}{d}.$$

Where in the last inequality we used $n \geq d$ to deduce $2\big(\frac{2en}{d}\big)^{-d} \leq \frac{12d}{n} \log \frac{2en}{d}$. $\qquad\square$

For any sample $S = \big((x_1, y_1), (x_2, y_2), \ldots (x_n, y_n)\big)$ and an index set $J \subseteq \{1, 2, \ldots n\}$, define the subsample $S_J = \big((x_{j_1}, y_{j_1}), (x_{j_2}, y_{j_2}), \ldots (x_{j_k}, y_{j_k})\big)$, where $J = \{j_i\}_{i=1}^{k}$ are the elements of $J$ ordered as $j_1 < j_2 \leq\, <\, j_k$.

**Lemma 9** (Upper bound of Theorem 4). *Let $\mathcal{H}$ be a concept class with VC dimension $d \geq 1$ and suppose $\eta < \frac{1}{4d}$. Then there exits a randomized learner $\mathcal{A} : \mathcal{Z}^\star \to [0,1]^{\mathcal{X}}$ such that for any distribution $\mathcal{D}$ on $\mathcal{X} \times \mathcal{Y}$ and all $n \geq 1/\eta$ we have:*

$$\lambda_n(\mathcal{A}|\mathcal{D}, \eta) \leq 4\sqrt{\eta d}\log\frac{e}{\eta d},$$

$$\texttt{excess}_{\mathcal{H},\mathcal{D}}(\mathcal{A}, n) \leq 32\sqrt{\eta d}\log\frac{e}{\eta d}, \text{ and}$$

$$\texttt{excess}_{\mathcal{H},\mathcal{D},\eta}(\mathcal{A}, n) \leq 36\sqrt{\eta d}\log\frac{e}{\eta d}.$$

*Proof.* Let $\eta \in (0, 1)$ and let $\mathcal{H}$ be a class of VC dimension $d$. The learner $\mathcal{A}$ is defined as follows: For a distribution $\mathcal{D}$ over $\mathcal{Z}$, $n \geq 2$, and an $n$-sample $S \sim \mathcal{D}^n$, let $S$ be a concatenation of samples $S_1$ and $S_2$ of sizes $n_1 = \lfloor n/2\rfloor$ and $n_2 = \lceil n/2\rceil$ respectively, and let $k = \lfloor\sqrt{d/4\eta}\rfloor$; note that by an easy computation, the condition $n \geq 1/\eta$ implies $k \leq n_1, n_2$, while the condition $\eta \leq \frac{1}{4d}$ implies $k \geq 1$. $\mathcal{A}$ samples a $k$-set $J \subseteq [n_1]$, uniformly at random. Then on the subsample $T = S_{1,J}$ it runs the learner $\mathcal{A}_{\mathcal{H}_T}$ from Proposition 6 using the concept class $\mathcal{H}_T \subseteq \mathcal{H}$ and the subsample $S_2$. That is, $\mathcal{A}(S) = \mathcal{A}_{\mathcal{H}_{T(S_1)}}(S_2)$.

It is easy to see that, by construction, $T \sim \mathcal{D}^k$. Hence, by Proposition 6, we can estimate the (clean) loss of $\mathcal{A}$ as

$$L_\mathcal{D}(\mathcal{A}, n) = \mathbb{E}_{T\sim\mathcal{D}^k} L_\mathcal{D}(\mathcal{A}_{\mathcal{H}_T}, n_2) \leq \mathbb{E}_{T\sim\mathcal{D}^k}\left(\inf_{h\in\mathcal{H}_T} L_\mathcal{D}(h) + \sqrt{\eta\log|\mathcal{H}_T|} + 4\sqrt{\frac{\log 2|\mathcal{H}_T|}{n_2}}\right)$$

$$\leq \mathbb{E}_{T\sim\mathcal{D}^k}\left(\inf_{h\in\mathcal{H}_T} L_\mathcal{D}(h) + 9\sqrt{\eta\log 2|\mathcal{H}_T|}\right).$$

Note that the last bound follows from the assumption $n_2 = \lceil n/2\rceil > \frac{1}{2\eta}$. We now separately bound the two terms in the above. The first term is bounded using Lemma 8:

$$\mathbb{E}_{T\sim\mathcal{D}^k}\inf_{h\in\mathcal{H}_T} L_\mathcal{D}(h) \leq [\text{by the definition of } \varepsilon\text{-cover}] \leq \inf_{h\in\mathcal{H}} L_\mathcal{D}(h) + \mathbb{E}_{T\sim\mathcal{D}^k}\mathcal{E}_\mathcal{D}(\mathcal{H}_T)$$

$$\leq \inf_{h\in\mathcal{H}} L_\mathcal{D}(h) + \frac{13d}{k}\log\frac{2ek}{d} \leq \inf_{h\in\mathcal{H}} L_\mathcal{D}(h) + 26\sqrt{\eta d}\log\frac{e}{\sqrt{\eta d}} \leq \inf_{h\in\mathcal{H}} L_\mathcal{D}(h) + 13\sqrt{\eta d}\log\frac{e^2}{\eta d}.$$

For the second term, we apply the Sauer–Shelah lemma 4 to conclude that $|\mathcal{H}_T| \leq (\frac{e\cdot k}{d})^d$. Note that the above bounds requires $k > d + 1$, which is enforced by the condition $\eta < \frac{1}{4d}$. Then

$$\mathbb{E}_{T\sim\mathcal{D}^k} 5\sqrt{\eta\log 2|\mathcal{H}_T|} \leq 5\sqrt{\eta d\log\frac{2ek}{d}} \leq 5\sqrt{\eta d}\log\frac{e}{\sqrt{\eta d}} \leq 3\sqrt{\eta d}\log\frac{e^2}{d\eta}.$$

Finally, combining these two estimates, we get:

$$L_\mathcal{D}(\mathcal{A}, n) \leq \inf_{h\in\mathcal{H}} L_\mathcal{D}(h)13\sqrt{\eta d}\log\frac{e^2}{\eta d} + 3\sqrt{\eta d}\log\frac{e^2}{d\eta} \leq \inf_{h\in\mathcal{H}} L_\mathcal{D}(h) + 32\sqrt{\eta d}\log\frac{e}{\eta d},$$

yielding

$$\texttt{excess}_{\mathcal{H},\mathcal{D}}(\mathcal{A}, n) \leq 32\sqrt{\eta d}\log\frac{e}{\eta d}.$$

as needed.

Now let us show the stability bound. Let $S, S' \in \mathcal{Z}^n$ be samples such that $S' \in B_\eta(S)$, let $x \in \mathcal{X}$ and, as above, let $J \subseteq [n_1]$, be a $k$-set, for $k = \lfloor\sqrt{d/4\eta}\rfloor$, chosen uniformly at random. Let $T = T(J) = S_{1,J}$ and $T' = T'(J) = S'_{1,J}$. Recall that the rendomness of $\mathcal{A}$ is composed of picking $J$ and the internal randomness $r$ of $\mathcal{A}_{\mathcal{H}_T}$. We have

$$\mathbb{P}_{J,r}\Big[\mathcal{A}(S)(x) \neq \mathcal{A}(S')(x)\Big] = \mathbb{P}_{J,r}\Big[\mathcal{A}_{\mathcal{H}_T}(S_2)(x) \neq \mathcal{A}_{\mathcal{H}_{T'}}(S'_2)(x)\Big]$$

$$\leq \mathbb{P}_J\Big[T(J) \neq T'(J)\Big] + \mathbb{P}_J\Big[T(J) = T'(J)\Big] \cdot \mathbb{P}_r\Big[\mathcal{A}_{\mathcal{H}_T}(S_2)(x) \neq \mathcal{A}_{\mathcal{H}_{T'}}(S'_2)(x)\,\Big|\, T = T'\Big] \leq \ldots$$

Let us estimate the first probability as follows: Let $I = \{i \in [n_1] \ : \ S_{1,i} \neq S'_{1,i}\}$. As $S' \in B_\eta(S)$, this implies $|I| \leq \eta \cdot n$, and so

$$\mathbb{P}_J\left[T(J) \neq T'(J)\right] = \mathbb{P}_J\left[S_{1,J} \neq S'_{1,J}\right] = \mathbb{P}_J\left[|J \cap I| \geq 1\right] \leq \mathbb{E}_J |J \cap I| \leq k\eta.$$

Thus

$$\ldots \leq k\eta + \mathbb{P}_J\left[T(J) = T'(J)\right] \cdot \mathbb{P}_r\left[\mathcal{A}_{\mathcal{H}_T}(S_2)(x) \neq \mathcal{A}_{\mathcal{H}_{T'}}(S'_2)(x) \ \Big| \ T = T'\right]$$

$$= k\eta + \mathbb{P}_J\left[T(J) = T'(J)\right] \cdot \mathbb{P}_r\left[\mathcal{A}_{\mathcal{H}_T}(S_2)(x) \neq \mathcal{A}_{\mathcal{H}_T}(S'_2)(x) \ \Big| \ T = T'\right]$$

$$\leq k\eta + \mathbb{P}_{J,r}\left[\mathcal{A}_{\mathcal{H}_T}(S_2)(x) \neq \mathcal{A}_{\mathcal{H}_T}(S'_2)(x)\right].$$

Notice that the only difference between the first and the second expression is that $\mathcal{A}_{\mathcal{H}_{T'}}$ becomes $\mathcal{A}_{\mathcal{H}_T}$. Now, taking expectation over $S \sim \mathcal{D}^n$ and $(x,y) \sim \mathcal{D}$ in the above, we get

$$\lambda_n(\mathcal{A}|\mathcal{D},\eta) \leq k\eta + \mathop{\mathbb{E}}_{\substack{S \sim \mathcal{D}^n \\ (x,y) \sim \mathcal{D}}} \sup_{S' \in B_\eta(S)} \mathbb{P}_{J,r}\left[\mathcal{A}_{\mathcal{H}_T}(S_2)(x) \neq \mathcal{A}_{\mathcal{H}_T}(S'_2)(x)\right]$$

$$\leq k\eta + \mathop{\mathbb{E}}_{\substack{S \sim \mathcal{D}^n \\ J \\ (x,y) \sim \mathcal{D}}} \sup_{S' \in B_\eta(S)} \mathbb{P}_r\left[\mathcal{A}_{\mathcal{H}_T}(S_2)(x) \neq \mathcal{A}_{\mathcal{H}_T}(S'_2)(x)\right]$$

$$\leq k\eta + \mathop{\mathbb{E}}_{T \sim \mathcal{D}^k} \mathop{\mathbb{E}}_{\substack{S_2 \sim \mathcal{D}^{n_2} \\ (x,y) \sim \mathcal{D}}} \sup_{S'_2 \in B_{2\eta}(S_2)} \mathbb{P}_r\left[\mathcal{A}_{\mathcal{H}_T}(S_2)(x) \neq \mathcal{A}_{\mathcal{H}_T}(S'_2)(x)\right]$$

$$\leq k\eta + \mathop{\mathbb{E}}_{T \sim \mathcal{D}^k} \lambda_{n_2}(\mathcal{A}_{\mathcal{H}_T}|\mathcal{D}, 2\eta)$$

$$\leq [\text{by Proposition 6}] \leq k\eta + \mathop{\mathbb{E}}_{T \sim \mathcal{D}^k} 4\sqrt{\eta \log |\mathcal{H}_T|}$$

$$\leq \left[\text{by } |\mathcal{H}_T| \leq \left(\frac{e \cdot k}{d}\right)^d\right] \leq k\eta + 4\sqrt{\eta d \log \frac{ek}{d}}$$

$$\leq \frac{\sqrt{\eta d}}{2} + 4\sqrt{\eta d} \log \frac{e}{\sqrt{4\eta d}} \leq \frac{\sqrt{\eta d}}{2} + 2\sqrt{\eta d} \log \frac{e^2}{4\eta d}$$

$$\leq 4\sqrt{\eta d} \log \frac{e}{\eta d}.$$

Finally, we estimate $\texttt{excess}_{\mathcal{H},\mathcal{D},\eta}(\mathcal{A}, n)$ by (1)

$$\texttt{excess}_{\mathcal{H},\mathcal{D},\eta}(\mathcal{A}, n) \leq \lambda_n(\mathcal{A}|\mathcal{D},\eta) + \texttt{excess}_{\mathcal{H},\mathcal{D}}(\mathcal{A}, n)$$

$$\leq 4\sqrt{\eta d} \log \frac{e}{\eta d} + 32\sqrt{\eta d} \log \frac{e}{\eta d} \leq 36\sqrt{\eta d} \log \frac{e}{\eta d}.$$

$\square$

### 5.1.2 Lower bound

When it comes to lower bounds on the excess error rate, it is convenient to consider *oblivious adversary*, which only changes the distribution, not the sample itself, and can be considered a relaxation of the regular sample-changing adversary. This setup can be informally summarized as follows:

1. The adversary selects a distribution $\mathcal{D}$.

2. A labeled example $z = (x, y)$ is drawn from $\mathcal{D}$ and is shown to the adversary.

3. The adversary corrupts the distribution $\mathcal{D}$ by changing it into $\mathcal{D}'$ which is $\eta$-close to $\mathcal{D}$.

4. The learner draws an $n$-sample $S$ from the corrupted distribution $\mathcal{D}'$ and uses it to predict the label $y$.

The formal definition follows Blanc and Valiant [2024]. Recall that for distributions $\mu_1$ over $X_1$ and $\mu_2$ over $X_2$, a *coupling* of $\mu_1$ and $\mu_2$ is a distribution $\mu_{12}$ over $X_1 \times X_2$, whose marginals on $X_1$ and $X_2$ coincide with $\mu_1$ and $\mu_2$ respectively. For distributions $\mathcal{D}_1$ and $\mathcal{D}_2$ over $\mathcal{Z}$, we define the distance between them as

$$\mathtt{d}(\mathcal{D}_1, \mathcal{D}_2) = \inf_{\mathcal{D}_{12}} \mathbb{P}_{z_1, z_2 \sim \mathcal{D}_{12}} [z_1 \neq z_2],$$

where the infimum is over all couplings $\mathcal{D}_{12}$ of $\mathcal{D}_1$ and $\mathcal{D}_2$. We note that thus defined, $\mathtt{d}(\mathcal{D}_1, \mathcal{D}_2)$ coincides with the *total variation distance* between $\mathcal{D}_1$ and $\mathcal{D}_2$, that is:

$$\mathtt{d}(\mathcal{D}_1, \mathcal{D}_2) = \sup_{E} |\mathcal{D}_1(E) - \mathcal{D}_2(E)|,$$

where the supremum is over all measurable events, see Definition 9 in Blanc and Valiant [2024]. Finally, for a distribution $\mathcal{D}$ we define $B_\eta(\mathcal{D})$ to be the set of all distributions $\mathcal{D}'$ such that $\mathtt{d}(\mathcal{D}, \mathcal{D}') \leq \eta$.

**Definition 10** ($\eta$-oblivious adversarial loss). *Let $\eta \in (0, 1)$ be the adversary's budget, let $\mathcal{A}$ be a (possibly randomized) learning rule, and let $\mathcal{D}$ be a distribution over examples. The $\eta$-oblivious adversarial loss of $\mathcal{A}$ with sample size $n$ and with respect to $\mathcal{D}$ is defined by*

$$L_{\mathcal{D},\eta}^{\mathtt{obl}}(\mathcal{A}, n) = \mathbb{E}_{(x,y)\sim\mathcal{D}} \Big[ \sup_{\mathcal{D}'\in B_\eta(\mathcal{D})} \mathbb{E}_{S'\sim\mathcal{D}'} \mathbb{E}_r |\mathcal{A}_r(S')(x) - y| \Big].$$

The following proposition substantiates the claim that the oblivious adversary can be considered a relaxation of the regular one.

**Proposition 11.** *For any $n > 0$, $\eta \in (0, 1)$, $\varepsilon > 0$, learner $\mathcal{A}$, and distribution $\mathcal{D}$ be we have*

$$L_{\mathcal{D},2\eta}(\mathcal{A}, n) + e^{\frac{-n\eta}{3}} \geq L_{\mathcal{D},\eta}^{\mathtt{obl}}(\mathcal{A}, n).$$

*Proof.* Recall that we want to upper-bound

$$L_{\mathcal{D},\eta}^{\mathtt{obl}}(\mathcal{A}, n) = \mathbb{E}_{(x,y)\sim\mathcal{D}} \Big[ \sup_{\mathcal{D}'\in B_\eta(\mathcal{D})} \mathbb{E}_{S'\sim\mathcal{D}'} \mathbb{E}_r |\mathcal{A}_r(S')(x) - y| \Big]$$

with

$$L_{\mathcal{D},\eta}(\mathcal{A}, n) = \mathbb{E}_{(x,y)\sim\mathcal{D}} \Big[ \mathbb{E}_{S\sim\mathcal{D}} \sup_{S'\in B_\eta(S)} \mathbb{E}_r |\mathcal{A}_r(S')(x) - y| \Big].$$

We will now bound the part after $\mathbb{E}_{x,y}$ of the first expression in terms of the similar part of the second. For brevity, we denote $\mathbb{E}_r |\mathcal{A}_r(S')(x) - y|$ by $L(S')$, keeping in mind that $L$ also depends on $x, y$, and $\mathcal{A}$.

$$\sup_{\mathcal{D}'\in B_\eta(\mathcal{D})} \mathbb{E}_{S'\sim\mathcal{D}'} L(S') = \sup_{\mathcal{D}'} \Big( \mathbb{P}_{\substack{S'\sim\mathcal{D}'\\S\sim\mathcal{D}}}[S' \in B_{2\eta}(S)] \cdot \mathbb{E}_{\substack{S'\sim\mathcal{D}',S\sim\mathcal{D}\\S'\in B_{2\eta}(S)}} L(S')$$

$$+ \mathbb{P}_{\substack{S'\sim\mathcal{D}'\\S\sim\mathcal{D}}}[S' \notin B_{2\eta}(S)] \cdot \mathbb{E}_{\substack{S'\sim\mathcal{D}',S\sim\mathcal{D}\\S'\notin B_{2\eta}(S)}} L(S') \Big) \leq \ldots$$

Here we assume that $\mathcal{D}$ and $\mathcal{D}'$ are coupled with a coupling witnessing $\mathtt{d}(\mathcal{D}, \mathcal{D}') \leq \eta$. Now, in the expression $[1] + [2]$ in brackets, let us estimate $[1]$ and $[2]$ separately, starting from $[2]$:

$$[2] = \mathbb{P}_{\substack{S'\sim\mathcal{D}'\\S\sim\mathcal{D}}}[S' \notin B_{2\eta}(S)] \cdot \mathbb{E}_{\substack{S'\sim\mathcal{D}',S\sim\mathcal{D}\\S'\notin B_{2\eta}(S)}} L(S')$$

$$\leq \mathbb{P}_{\substack{S'\sim\mathcal{D}'\\S\sim\mathcal{D}}}[S' \notin B_{2\eta}(S)] \leq \mathbb{P}[\mathrm{Bin}(n, \eta) \geq 2\eta n))] \leq e^{-\frac{n\eta}{3}}.$$

Here in the second inequality we use the fact that $\mathcal{D}$ and $\mathcal{D}'$ are coupled in such a way that $\mathbb{P}_{z\sim\mathcal{D},z'\sim D'}[z \neq z'] \leq \eta$, and hence the event $S' \notin B_{2\eta}(S)$ is equivalent to making at least

$2\eta n$ mistakes in $n$ trials, where the probability of an individual mistake is at most $\eta$. The final bound is by multiplicative Chernoff bound inequality. Now, for [1]:

$$[1] = \underset{\substack{S'\sim\mathcal{D}' \\ S\sim\mathcal{D}}}{\mathbb{P}}[S' \in B_{2\eta}(S)] \cdot \underset{\substack{S'\sim\mathcal{D}',S\sim\mathcal{D} \\ S'\in B_{2\eta}(S)}}{\mathbb{E}} L(S')$$

$$\leq \underset{\substack{S'\sim\mathcal{D}' \\ S\sim\mathcal{D}}}{\mathbb{P}}[S' \in B_{2\eta}(S)] \cdot \underset{\substack{S'\sim\mathcal{D}',S\sim\mathcal{D} \\ S'\in B_{2\eta}(S)}}{\mathbb{E}} \underset{S''\in B_{2\eta}(S)}{\sup} L(S'')$$

$$\leq \underset{S'\sim\mathcal{D}',S\sim\mathcal{D}}{\mathbb{E}} \underset{S''\in B_{2\eta}(S)}{\sup} L(S'') = \underset{S\sim\mathcal{D}}{\mathbb{E}} \underset{S'\in B_{2\eta}(S)}{\sup} L(S').$$

Note that here we fold back the expectation with conditionals, in the way opposite to how we did it in the beginning, but after changing the function under the expectation. Notably, the estimates for both [1] and [2] no longer depend on $\mathcal{D}'$, so

$$\ldots \leq \underset{S\sim\mathcal{D}}{\mathbb{E}} \underset{S'\in B_{2\eta}(S)}{\sup} L(S') + e^{\frac{-n\eta}{3}},$$

yielding the desired bound. $\qquad\square$

Throughout the section, we are going to assume the following setup. Our label space will be $\{\pm 1\}$ instead of the usual $\{0, 1\}$. Let $\mathcal{H} \subseteq \{\pm 1\}^{\mathcal{X}}$ be a concept class of VC dimension $d$, and let $X = \{x_i\}_{i=1}^d$ be a fixed $d$-set shattered by $\mathcal{X}$. We will only consider distributions that have uniform marginals over $\{x_i\}_{i=1}^d$, that is, distributions $\mathcal{D}$ such that for all $i \leq 1 \leq d$ we have

$$\underset{(x,y)\sim\mathcal{D}}{\mathbb{P}}[x = x_i] = \frac{1}{d}.$$

For $u \in I^d$, let us define a distribution $\mathcal{D}_u$ as

$$\underset{(x,y)\sim D_u}{\mathbb{P}}\left[(x,y) = (x_i, y)\right] = \frac{1}{d}(1/2 + yu_i) = \begin{cases} \frac{1}{d}\left(\frac{1}{2} + u_i\right) & y = 1, \\ \frac{1}{d}\left(\frac{1}{2} - u_i\right) & y = -1. \end{cases}$$

Note that the function $u \mapsto \mathcal{D}_u$ gives a one to one correspondence between distributions with uniform marginals and $[-\frac{1}{2}, \frac{1}{2}]^d$, which we are going to denote by $I^d = [-\frac{1}{2}, \frac{1}{2}]^d$. Moreover, if we define the metric on $I^d$ as $l_1$-norm rescaled by $1/d$, that is,

$$\mathtt{d}(u, u') = \frac{1}{d}\|u - u'\|_1 = \frac{1}{d}\sum_{i=1}^d |u_i - u'_i|,$$

then $\mathtt{d}(u, u') = \mathtt{d}(\mathcal{D}_u, \mathcal{D}'_u)$.

For a fixed $n$ and a learner $\mathcal{A} : \mathcal{Z}^\star \to [0,1]^{\mathcal{X}}$, let us define a function $F = F(\mathcal{A}, n) : I^d \to I^d$ as

$$F_i(u) = \frac{1}{2} \underset{S\sim\mathcal{D}^n}{\mathbb{E}} \mathcal{A}(S)(x_i).$$

Now, for an *arbitrary* function $F : I^d \to I^d$, let us define the *$\eta$-oblivious adversarial loss* of $F$ and *$\eta$-excess* of $F$ with respect to $u \in I^d$ as

$$L_{u,\eta}^{\mathtt{obl}}(F) = \frac{1}{d} \sum_{\substack{i=1,\ldots,d \\ y\in\{\pm 1\}}} \underset{u'\in B_\eta(u)}{\sup} (1/2 + yu_i)(1/2 - yF_i(u')), \text{ and}$$

$$\mathtt{excess}_{u,\eta}^{\mathtt{obl}}(F) = L_{u,\eta}^{\mathtt{obl}}(F) - \frac{1}{d} \sum_{i=1,\ldots,d} \min(1/2 - u_i, 1/2 + u_i).$$

In particular, for $F = F(\mathcal{A}, n)$, we have:

$$L^{\text{obl}}_{\mathcal{D}_u,\eta}(\mathcal{A}, n) = \mathop{\mathbb{E}}_{(x,y)\sim\mathcal{D}_u}\left[\sup_{u'\in B_\eta(u)}\mathop{\mathbb{E}}_{S'\sim\mathcal{D}_{u'}}\mathop{\mathbb{E}}_{r}\frac{|\mathcal{A}_r(S')(x) - y|}{2}\right]$$

$$= \frac{1}{d}\sum_{\substack{i=1,\ldots,d\\y\in\{\pm 1\}}}(1/2 + yu_i)\cdot\left[\sup_{u'\in B_\eta(u)}\mathop{\mathbb{E}}_{S'\sim\mathcal{D}_{u'}}\mathop{\mathbb{E}}_{r}\frac{|\mathcal{A}_r(S')(x) - y|}{2}\right]$$

$$= \frac{1}{d}\sum_{\substack{i=1,\ldots,d\\y\in\{\pm 1\}}}\left[\sup_{u'\in B_\eta(u)}(1/2 + yu_i)\cdot\left(1/2 - y\cdot 1/2\cdot\mathop{\mathbb{E}}_{S'\sim\mathcal{D}_{u'}}\mathop{\mathbb{E}}_{r}\mathcal{A}_r(S')(x_i)\right)\right]$$

$$= \frac{1}{d}\sum_{\substack{i=1,\ldots,d\\y\in\{\pm 1\}}}\sup_{u'\in B_\eta(u)}(1/2 + yu_i)\left(1/2 - yF_i(u')\right) = L^{\text{obl}}_{u,\eta}(F(\mathcal{A}, n)),$$

and, similarly,

$$\text{excess}^{\text{obl}}_{\mathcal{D}_u,\eta}(\mathcal{A}, n) = \text{excess}^{\text{obl}}_{u,\eta}(F(\mathcal{A}, n)). \tag{2}$$

Note that the $1/2$ factor in the first equality for the loss comes from changing the label space from $\{0, 1\}$ to $\{\pm 1\}$.

A poisoning scheme is a collection $\xi = \{\xi_{i,y}\}_{i\in[d],y\in\{\pm 1\}}$ of functions from $I^d$ to $I^d$. The scheme $\xi$ is said to have a poisoning budget $\eta$ if $\mathsf{d}(\xi_{i,y}(u), u) = 1/d\cdot\|\xi_{i,y}(u) - u\|_1 \leq \eta$ for all $i \in [d]$, $y \in \{\pm 1\}$, and $u \in I^d$. Poisoning schemes are used to explicate the choice of $u' \in B_\eta(u)$, and so we define $L^{\text{obl}}_{u,\xi}(F)$ and $\text{excess}^{\text{obl}}_{u,\xi}(F)$ by tweaking, in an obvious way, the respective definitions of $L^{\text{obl}}_{u,\eta}(F)$ and $\text{excess}^{\text{obl}}_{u,\eta}(F)$. Trivially, for all $F: I^d \to I^d$ and $u \in I^d$, we have

$$\text{excess}^{\text{obl}}_{u,\eta}(F) = \sup_\xi\text{excess}^{\text{obl}}_{u,\xi}(F),$$

where the infimum is over all poisoning schemes with budget $\eta$. We thus want to prove that for every such $F$ there is $u \in I^d$ and a poisoning scheme $\xi$ with budget $\eta$ such that

$$\text{excess}^{\text{obl}}_{u,\xi}(F) = \Omega(\sqrt{d\eta}).$$

**Lemma 12** (One-dimensional lower bound). *For any $\eta \in (0, 1)$, there exist a (one dimensional) poisoning scheme $\xi = (\xi_{-1}, \xi_1)$ with poisoning budget $\eta$ and a distribution $\mathcal{U}$ over $I$ such that for any $F : I \to I$ we have*

$$\mathop{\mathbb{E}}_{u\sim\mathcal{U}}\text{excess}^{\text{obl}}_{u,\xi}(F) \geq \frac{\sqrt{\eta}}{16}.$$

*Proof.* We are going to prove that such $\xi$ and $\mathcal{U}$ exist for $\eta \leq 1/16$, and that they guarantee $E_\eta = \mathbb{E}_{u\sim\mathcal{U}}\text{excess}^{\text{obl}}_{u,\xi}(F) \geq \frac{\sqrt{\eta}}{4}$. Note that in this case, for $\eta \geq 1/16$ (but still $\eta \leq 1$), we can thus enforce an excess of $E_{1/16} = \sqrt{1/16}/4 = 1/16$, yielding the statement of the lemma for all $\eta \in (0, 1)$.

So let $\eta \leq 1/16$ and let $m$ be a natural number such that $\sqrt{\eta}/2 \leq (2m + 1)\eta \leq \sqrt{\eta} \leq 1/2$; it is easy to see that the condition on $\eta$ guarantees that such $m$ exists. Let $U = \{2i\eta : -m \leq i \leq m\}$. Define the poisoning scheme $(\xi_{-1}, \xi_1)$ as

$$\xi_{-1}(2i\eta) = (2i + 1)\eta,$$
$$\xi_1(2i\eta) = (2i - 1)\eta,$$

for $i = -m, \ldots, m$ and $\xi_0(x) = \xi_1(x) = x$ for $x \in I$, $x \neq 2i\eta$. Clearly, the poisoning budget of $\xi = (\xi_0, \xi_1)$ is $\eta$. Let us now estimate the excess error for this poisoning scheme first at points $2i\eta$, for $i = -m, \ldots, m$, and then, in a different way, at points $-(2m + 1)\eta$ and $(2m + 1)\eta$. So, for $0 \leq i \leq m$, and $u = 2i\eta$:

$$\text{excess}^{\text{obl}}_{u,\xi}(F) = \sum_{y\in\{\pm 1\}}(1/2 + yu)\left(1/2 - yF(\xi_y(u))\right) - (1/2 - u)$$

$$= (1/2 + u)\left(1/2 - F(\xi_1(u))\right) + (1/2 - u)\left(1/2 + F(\xi_{-1}(u))\right) - (1/2 - u)$$

$$= \frac{F(u + \eta) - F(u - \eta)}{2} + u\left(1 - F(u - \eta) - F(u + \eta)\right).$$

By a similar computation, for $-m \leq i \leq 0$ and $u = 2i\eta$:

$$\texttt{excess}^{\texttt{obl}}_{u,\xi}(F) = \frac{F(u+\eta) - F(u-\eta)}{2} - u\big(1 + F(u-\eta) + F(u+\eta)\big).$$

In either case, for $u = 2i\eta$ and $i = -m, \ldots, m$, we have:

$$\texttt{excess}^{\texttt{obl}}_{u,\xi}(F) \geq \frac{F(u+\eta) - F(u-\eta)}{2}. \tag{3}$$

At the same time, for $u = (2m+1)\eta$, $\xi_{-1}(u) = \xi_1(u) = u$, and so

$$\begin{aligned}
\texttt{excess}^{\texttt{obl}}_{u,\xi}(F) &= \sum_{y \in \{\pm 1\}} (1/2 + yu)\big(1/2 - yF(u)\big) - (1/2 - u) \\
&= (1/2 + u)\big(1/2 - F(u)\big) + (1/2 - u)\big(1/2 + F(u)\big) - (1/2 - u) \\
&= \big[1/2 + F(u) = 1 - \big(1/2 - F(u)\big)\big] \\
&= (1/2 + u)\big(1/2 - F(u)\big) - (1/2 - u)\big(1/2 - F(u)\big) \\
&= u\big(1 - 2F(u)\big).
\end{aligned}$$

Similarly, for $u = -(2m+1)\eta$,

$$\texttt{excess}^{\texttt{obl}}_{u,\xi}(F) = -u\big(1 + 2F(u)\big).$$

All in all, for $u = (2m+1)\eta$, we get

$$\texttt{excess}^{\texttt{obl}}_{u,\xi}(F) = |u|\big(1 - 2\mathsf{sign}(u)F(u)\big). \tag{4}$$

Let us now define the tistribution $\mathcal{U}$ as follows: let $\mathcal{U}_1$ be a uniform distribution over $U = \{2i\eta : -m \leq i \leq m\}$, let $\mathcal{U}_2$ be a uniform distribution over $\{\pm(2m+1)\eta\}$, and let $\mathcal{U}$ be $U_1$ or $\mathcal{U}_2$ with probability $1/2$ each. Trivially,

$$\mathop{\mathbb{E}}_{u \sim \mathcal{U}} \texttt{excess}^{\texttt{obl}}_{u,\xi}(F) = \frac{1}{2} \mathop{\mathbb{E}}_{u \sim \mathcal{U}_1} \texttt{excess}^{\texttt{obl}}_{u,\xi}(F) + \frac{1}{2} \mathop{\mathbb{E}}_{u \sim \mathcal{U}_2} \texttt{excess}^{\texttt{obl}}_{u,\xi}(F).$$

Let us estimate the two terms in the above using (3) and (4). Below, we denote $(2m+1)\eta$ by $t$:

$$\begin{aligned}
\mathop{\mathbb{E}}_{u \sim \mathcal{U}_1} \texttt{excess}^{\texttt{obl}}_{u,\xi}(F) &\geq [\text{by (3)}] \geq \mathop{\mathbb{E}}_{u \sim \mathcal{U}} \frac{F(u+\eta) - F(u-\eta)}{2} \\
&= \frac{1}{2m+1} \sum_{k=-m}^{m} \frac{F(2k\eta + \eta) - F(2k\eta - \eta)}{2} = \frac{F(t) - F(-t)}{4m+2} = \eta \frac{F(t) - F(-t)}{2t}.
\end{aligned}$$

and

$$\begin{aligned}
\mathop{\mathbb{E}}_{u \sim \mathcal{U}_2} \texttt{excess}^{\texttt{obl}}_{u,\xi}(F) &\geq [\text{by (4)}] \geq \frac{1}{2}t(1 - 2F(t)) + \frac{1}{2}t(1 + 2F(-t)) \\
&= t - t(F(t) - F(-t)).
\end{aligned}$$

Combining, we get

$$\begin{aligned}
\mathop{\mathbb{E}}_{u \sim \mathcal{U}} \texttt{excess}^{\texttt{obl}}_{u,\xi}(F) &= \frac{1}{2}\left(t - t\big(F(t) - F(-t)\big) + \eta \frac{F(t) - F(-t)}{2t}\right) \\
&= \frac{t}{2} + \big(F(t) - F(-t)\big)\frac{\eta - 2t^2}{4t} \geq \frac{t}{2} - \left|\frac{2t^2 - \eta}{4t}\right| \\
&\geq \left[\sqrt{\eta} \geq t \geq \frac{\sqrt{\eta}}{2}\right] \geq \frac{t}{2} - \frac{\eta}{4t} \geq \frac{\eta}{4t} \geq \frac{\sqrt{\eta}}{4}.
\end{aligned}$$

$\square$

**Lemma 13** (Lower bound for $F \colon I^d \to I^d$)**.** *Let $\eta$ be a poisoning budget with $\eta < 1/d$. There exists a distribution $\mathcal{U}$ over $I$ and a poisoning scheme $\xi$ with budget $\eta$ such that for any function $F \colon I^d \to I^d$ we have*

$$\mathop{\mathbb{E}}_{u \sim \mathcal{U}^d} \mathtt{excess}^{\mathtt{obl}}_{u,\xi}(F) \geq \frac{\sqrt{d\eta}}{16}.$$

*Proof.* By Lemma 12, there exists a distribution $\mathcal{U}$ over $I$ and a poisoning scheme $\hat\xi = (\hat\xi_0, \hat\xi_1)$ with poisoning budget $d\eta$ such that for any function $F \colon I \to I$ we have

$$\mathop{\mathbb{E}}_{u \sim \mathcal{U}} \mathtt{excess}^{\mathtt{obl}}_{u,\hat\xi}(F) \geq \frac{\sqrt{d\eta}}{16}.$$

Note that here we need $d\eta < 1$, which is enforced by our condition on $\eta$. Let us define a $d$-dimensional poisoning strategy $\xi = \{(\xi_{i,0}, \xi_{i,1})\}_{i=1}^d$ as

$$\xi_{i,y}(u) = \big(u_1, \ldots u_{i-1}, \hat\xi_y(u_i), u_{i+1}, \ldots, u_d\big).$$

Note that since the budget of $\hat\xi$ is $d\eta$, the poisoning budget of $\xi$ is $\eta$. Now, for a given $G \colon I^d \to I^d$, let us define $F = F(G) \colon I \to I$ as

$$F(t) = \mathop{\mathbb{E}}_{u \sim \mathcal{U}^d} \frac{1}{d} \sum_{i=1}^d G_i(u_1, \ldots, u_{i-1}, t, u_{i+1}, \ldots, u_d) = \frac{1}{d} \sum_{i=1}^d F_i(t),$$

where $F_i(t) = \mathbb{E}_{u \sim \mathcal{U}^d} G_i(u_1, \ldots, u_{i-1}, t, u_{i+1}, \ldots, u_d)$.

In words, $F(t)$ is an expected value of $G_i(\ldots, t, \ldots)$, where $i$ is one of the $d$ dimensions picked at random, $t$ is plugged into the $i$'th coordinate, and the rest of coordinates a picked at random according to $\mathcal{U}$. Moreover, by design, the expected excess error of $F$ with $\hat\xi$ is the same as of $G$ with $\xi$, but let us nevertheless explicate it. In what follows, let $\min_{u_i} := \min(1/2 - u_i, 1/2 + u_i)$.

$$\mathop{\mathbb{E}}_{u \sim \mathcal{U}^d} \mathtt{excess}^{\mathtt{obl}}_{u,\xi}(G) = \mathop{\mathbb{E}}_{u \sim \mathcal{U}^d} \frac{1}{d} \sum_{i=1,\ldots,d} \left[ \sum_{y \in \{\pm 1\}} (1/2 + yu_i)\big(1/2 - yG_i(\xi_{i,y}(u))\big) - \min_{u_i} \right]$$

$$= \left[ \text{drag} \mathop{\mathbb{E}}_{u_j \sim \mathcal{U}}, \text{ for } j \neq i, \text{ inside, all the way down to } G_i \right]$$

$$= \frac{1}{d} \sum_{i=1,\ldots,d} \mathop{\mathbb{E}}_{u_i \sim \mathcal{U}} \left[ \sum_{y \in \{\pm 1\}} (1/2 + yu_i) \left( 1/2 - y \cdot \mathop{\mathbb{E}}_{\substack{u_j \sim \mathcal{U}^{d-1} \\ j \in [d]-i}} G_i(\xi_{i,y}(u)) \right) - \min_{u_i} \right]$$

$$= \left[ \text{rename } u_i \text{ to } t \text{ and rewrite using } \mathop{\mathbb{E}}_{u \sim \mathcal{U}^d} G_i(\xi_{i,y}(u)) = F_i(\hat\xi_y(u)) \right]$$

$$= \frac{1}{d} \sum_{i=1,\ldots,d} \mathop{\mathbb{E}}_{t \sim \mathcal{U}} \left[ \sum_{y \in \{\pm 1\}} (1/2 + yt) \left( 1/2 - y \cdot F_i(\hat\xi_y(t)) \right) - \min_t \right]$$

$$= \left[ \text{now drag the average over } i \text{ all the way down to } F_i, \text{ and use } F(t) = \frac{1}{d} \sum_i F_i(t) \right]$$

$$= \mathop{\mathbb{E}}_{t \sim \mathcal{U}} \left[ \sum_{y \in \{\pm 1\}} (1/2 + yt) \left( 1/2 - y \cdot F(\hat\xi_y(t)) \right) - \min_t \right] = \mathop{\mathbb{E}}_{t \sim \mathcal{U}} \mathtt{excess}^{\mathtt{obl}}_{t,\hat\xi}(F).$$

In particular, this implies

$$\mathop{\mathbb{E}}_{u \sim \mathcal{U}^d} \mathtt{excess}^{\mathtt{obl}}_{u,\xi}(G) \geq \frac{\sqrt{d\eta}}{16},$$

as needed. $\qquad\qquad\square$

**Lemma 14** (Lower bound of Theorem 4). *For any concept class $\mathcal{H}$ of VC dimension $d \geq 1$, any randomized learner $\mathcal{A} : (\mathcal{X} \times \mathcal{Y})^\star \to [0,1]^{\mathcal{X}}$, poisoning budget $\eta < 1/d$ and sample size $n \geq \frac{6}{\eta} \log \frac{64}{\sqrt{d\eta}}$, there exists a distribution $\mathcal{D}$ such that*

$$\mathtt{excess}_{\mathcal{H},\mathcal{D},\eta}(\mathcal{A}, n) \geq \inf_{h \in \mathcal{H}} L_{\mathcal{D}_u}(h) + \frac{\sqrt{d\eta}}{16\sqrt{2}} - e^{\frac{-\eta n}{6}}.$$

*In particular, for $n > \frac{6}{\eta} \log \frac{64}{\sqrt{d\eta}}$ we have*

$$\mathtt{excess}_{\mathcal{H},\mathcal{D},\eta}(\mathcal{A}, n) \geq \inf_{h \in \mathcal{H}} L_{\mathcal{D}_u}(h) + \frac{\sqrt{d\eta}}{36}.$$

*Proof.* Recall that Lemma 13, applied to $F = F(\mathcal{A}, n) \colon I^d \to I^d$, for an arbitrary sample size $n$, implies that there is a distribution $\mathcal{D} = \mathcal{D}_u$, realizable by $\mathcal{H}$ and corresponding to $u \in I^d$, and a poisoning scheme $\xi$ with budget $\frac{\eta}{2}$ such that

$$\mathtt{excess}^{\mathrm{obl}}_{\mathcal{D},\eta/2}(\mathcal{A}, n) = \mathtt{excess}^{\mathrm{obl}}_{u,\eta/2}(F(\mathcal{A}, n)) \geq \mathtt{excess}^{\mathrm{obl}}_{u,\xi}(F(\mathcal{A}, n)) \geq \frac{\sqrt{d\eta}}{16\sqrt{2}}.$$

Here, the first equality is by (2), the second inequality is by the definition of poisoning scheme with budget, and the last one is by Lemma 13, where we change a probabilistic statement into an existential one. Note that here we use the condition $\eta < \frac{1}{d}$ for compliance with the lemma.

Finally, exchanging the oblivious poisoning with a regular one by Proposition 11, we get:

$$\mathtt{excess}_{\mathcal{D},\eta}(\mathcal{A}, n) \geq \mathtt{excess}^{\mathrm{obl}}_{\mathcal{D},\eta/2}(\mathcal{A}, n) - e^{\frac{-n\eta/2}{3}} \geq \frac{\sqrt{d\eta}}{16\sqrt{2}} - e^{\frac{-n\eta}{6}}.$$

Which gives the first assertion of the Lemma. For the second a simple computation shows that for $n \geq \frac{6}{\eta} \log \frac{64}{\sqrt{d\eta}}$ we have

$$\frac{\sqrt{d\eta}}{16\sqrt{2}} - e^{\frac{-\eta n}{6}} \geq \frac{\sqrt{d\eta}}{16\sqrt{2}} - \exp(\log \frac{-64}{\sqrt{d\eta}}) = \sqrt{d\eta}(\frac{1}{16\sqrt{2}} - \frac{1}{64}) \geq \frac{\sqrt{d\eta}}{36}.$$

$\square$

## 5.2 Proofs of Theorems 2 and 3

**Theorem 2.** *For every learning rule $\mathcal{A}_{\mathtt{priv}}$, there exists a learning rule $\mathcal{A}_{\mathtt{pub}}$ such that for any distribution $\mathcal{D}$, sample size $n$, and poisoning budget $\eta \in (0,1)$,*

$$L^{\mathtt{pub}}_{\mathcal{D},\eta}(\mathcal{A}_{\mathtt{pub}}, n) \leq L_{\mathcal{D},\eta}(\mathcal{A}_{\mathtt{priv}}, n).$$

*Moreover, $\mathcal{A}_{\mathtt{pub}}$ can be constructed efficiently given black-box access to $\mathcal{A}_{\mathtt{priv}}$.*

Note that originally Theorem 2 was stated in terms of excess error; its equivalence with the present statement in terms of loss is straightforward.

*Proof.* In the proof, without losing generality, we identify the probability space of $\mathcal{A}_{\mathtt{priv}}$ and $\mathcal{A}_{\mathtt{pub}}$ with $[0,1]$. Let us define a function $A : \mathcal{Z}^\star \to [0,1]^{\mathcal{X}}$, induced by $\mathcal{A}_{\mathtt{priv}}$, as

$$A(S)(x) = \mathop{\mathbb{E}}_{r} \mathcal{A}_{\mathtt{priv},r}(x).$$

And let $\mathcal{A}_{\mathtt{pub},r}(S)(x) = 1$ if $r \leq A(S)$ and 0 otherwise. Let $\eta \in (0,1)$, and let $\mathcal{D}$ be some distribution. For each $x \in \mathcal{X}$ let us define functions $\xi_{x,0}, \xi_{x,1} : \mathcal{Z}^\star \to \mathcal{Z}^\star$ by

$$\xi_{x,0}(S) = \arg \max_{S' \in B_\eta(S)} A(S')(x),$$

$$\xi_{x,1}(S) = \arg \min_{S' \in B_\eta(S)} A(S')(x).$$

Note that, trivially, for $y \in \{0,1\}$, $\xi_{x,y}(S) \in B_\eta(S)$ and

$$\xi_{x,y}(S) = \arg \max_{S' \in B_\eta(S)} \mathbb{E}_r |\mathcal{A}_{\mathtt{priv},r}(S')(x) - y|.$$

We claim that $\mathcal{A}_{\mathtt{pub},r}\big(\xi_{x,y}(S)(x)\big) = y$ implies $\mathcal{A}_r(S')(x) = y$ for all $S' \in B_\eta(S)$. Indeed

$$\mathcal{A}_{\mathtt{pub},r}\big(\xi_{x,1}(S)(x)\big) = 1 \implies [\forall S' \in B_\eta(S)\,,\ r \leq A(S')] \implies [\forall S' \in B_\eta(S)\,,\ \mathcal{A}_{\mathtt{pub},r}(S)(x) = 1],$$
$$\mathcal{A}_{\mathtt{pub},r}\big(\xi_{x,0}(S)(x)\big) = 0 \implies [\forall S' \in B_\eta(S)\,,\ r > A(S')] \implies [\forall S' \in B_\eta(S)\,,\ \mathcal{A}_{\mathtt{pub},r}(S)(x) = 0].$$

Hence we have

$$L_{\mathcal{D},\eta}(\mathcal{A}_{\mathtt{priv}}, n) = \mathop{\mathbb{E}}_{\substack{S \sim \mathcal{D}^n \\ (x,y) \sim \mathcal{D}}} \Big[ \sup_{S' \in B_\eta(S)} \mathbb{E}_r |\mathcal{A}_{\mathtt{priv},r}(S')(x) - y| \Big] = \mathop{\mathbb{E}}_{\substack{S \sim \mathcal{D}^n \\ (x,y) \sim \mathcal{D}}} \Big[ \mathbb{E}_r |\mathcal{A}_{\mathtt{priv},r}\big(\xi_{x,y}(S)\big)(x) - y| \Big]$$

$$= \mathop{\mathbb{E}}_{\substack{S \sim \mathcal{D}^n \\ (x,y) \sim \mathcal{D}}} \big[ |\mathcal{A}\big(\xi_{x,y}(S)\big)(x) - y| \big] = \mathop{\mathbb{E}}_{\substack{S \sim \mathcal{D}^n \\ (x,y) \sim \mathcal{D}}} \mathbb{E}_r \Big[ \sup_{S' \in B_\eta(S)} |\mathcal{A}_{\mathtt{pub},r}(S')(x) - y| \Big] = L_{\mathcal{D},\eta}^{\mathtt{pub}}(\mathcal{A}_{\mathtt{pub}}, n).$$

$\square$

**Theorem 3** (Poisoned Learning Curves). *Let $\mathcal{H}$ be a concept class with VC dimension $d$, and let $\eta \in (0,1)$ be the poisoning budget. Then, for every learning rule $\mathcal{A}$, there exists a distribution $\mathcal{D}$ and an adversary that forces an excess error of at least $\Omega\big(\min\{\sqrt{d\eta}, 1\}\big)$ for infinitely many sample sizes $n$.*

*Proof.* The proof is by careful examination of the proof of Theorem 4, more precisely, of Lemmas 13 and 14. For the rest of the proof assume that $\eta < \frac{1}{d}$, otherwise the result is obvious.

First, we note that in Lemma 13, the parameter $u \in I^d$ of the target distribution $\mathcal{D}_u$, demonstrating the high excess of $F_n = F(\mathcal{A}, n)$, is picked from a distribution $\mathcal{U}$ on $I^d$, which has a finite support and whose construction is independent on $n$. Thus, by pigeonhole principle, there is $u \in I^d$ such that for $\mathcal{D} = \mathcal{D}_u$ and for infinitely many $n$

$$\mathtt{excess}_{\mathcal{D},\nu}^{\mathtt{obl}}(\mathcal{A}, n) = \mathtt{excess}_{u,\nu}^{\mathtt{obl}}(F(\mathcal{A}, n)) \geq \frac{\sqrt{d\eta}}{16}.$$

Which by Proposition 11 implies

$$\mathtt{excess}_{\mathcal{D},\eta}(\mathcal{A}, n) \geq \mathtt{excess}_{\mathcal{D},\eta/2}^{\mathtt{obl}}(\mathcal{A}, n) - e^{\frac{-n\eta/2}{3}} \geq \frac{\sqrt{d\eta}}{16\sqrt{2}} - e^{\frac{-n\eta}{6}}.$$

And since $\frac{\sqrt{d\eta}}{16} - e^{\frac{-n\eta}{6}} \leq \frac{\sqrt{d\eta}}{36}$ for only finitely many $n$'s, we deduce that for infinitely many $n$

$$\mathtt{excess}_{\mathcal{D},\eta}(\mathcal{A}, n) \geq \frac{\sqrt{d\eta}}{36}.$$

$\square$

