# OpenReview forum: "Agnostic Learning under Targeted Poisoning: Optimal Rates and the Role of Randomness"
_NeurIPS.cc/2025/Conference — NeurIPS 2025 spotlight_

### Official Review · Reviewer_yng4 · 2025-06-28

**Clarity:** 3
**Significance:** 2
**Originality:** 3
**Rating:** 4
**Confidence:** 4

**Summary:**

The setting

The paper studies PAC learning in the targeted poisoning model, building on [Hanake et al 2022] and preceding work. This setting is defined as follows:
- As in standard PAC learning, there is some hypothesis class $\mathcal{H}$.
- The adversary receives a collection $S$ of i.i.d. labeled samples from a distribution $D$, and has the power to corrupt $\eta$ fraction of elements in $S$.
- As in standard PAC learning, in the **realizable setting**, the original dataset $S$ is labeled by a function in $\mathcal{H}$. In the **agnostic setting** this is not the case, and the best error in $\mathcal{H}$ is used as a benchmark.
- The goal of the adversary is to force the learning algorithm to misclassify some specific point $x_0$, which is **drawn in the very beginning** from the sample distribution $D$ and given to the adversary.

For a specific choice of learning algorithm $\mathcal{A}$ and an adversary, the **adversarial loss** in this model is defined to be the probability (over all random variables including $x_0$) that the learning algorithm indeed misclassifies $x_0$ (and thus the adversary succeeds). Overall, this setting models the situation when the adversary wants the learning algorithm to misclassify some specific input $x_0$.

There are two distinctions that are important to keep in mind:
- As mentioned earlier, the setting can be **realizable** and **agnostic**.
- The learning algorithm $\mathcal{A}$ can be **deterministic** or it can be **randomized**. A deterministic learner outputs a classifier based only on the training dataset,  whereas a randomized learner can utilize additional random bits in addition to the training dataset.

It is slightly subtle how the goal of the learning algorithm is defined in the agnostic setting: the goal is to bound the **excess loss**, which is defined as the difference between (a) the **adversarial loss** defined above and (b) the minimum prediction error OPT in the class $\mathcal{H}$ on the distribution of $D$ from which data is drawn prior to being poisoned by the adversary.

Results

The paper focuses on the **agnostic** setting and allows the learning algorithm to be **randomized**.
If $d$ denotes the VC dimension of the class $\mathcal{H}$, the paper gives upper and lower bounds of $\tilde{\Theta}(\sqrt{d\eta})$ for the excess loss in this setting. In other words, the paper shows that there is a learning algorithm that achieves excess loss of  $\tilde{O}(\sqrt{d\eta})$, and this amount of loss is provably unavoidable.
Previous work [Hanake et al 2022] gave a bound of $\Theta(d\eta)$ for the realizable setting.
This fully characterizes the realizable setting, leaving open only the agnostic setting, in which:
- The work of [Hanake et al 2022] gave a bound on the adversarial loss of $C\cdot OPT+\Theta(d\eta)$ where $C$ is an absolute constant. [Hanake et al 2022] show that a multiplicative constant $C$ is necessary here, as long as a **deterministic** learner is used.
- As explained earlier, this work gives an algorithm with adversarial loss by $OPT+\Theta(\sqrt{d\eta})$. The impossibility result of [Hanake et al 2022] is side-stepped by using a randomized learning algorithm.

Additionally,  the upper bound given in this work is shown to work even if the adversary knows in advance the random bits utilized by the learning algorithm.
The work also gives lower bound instances that force excess error $\tilde{\Omega}(\sqrt{d\eta})$ for infinitely many dataset sizes $n$ (as opposed to having a different instance for each $n$).

Technical summary

The algorithm given in this work samples a classifier randomly from via a procedure inspired by the exponential mechanism. To handle continuous hypothesis classes, the paper uses a covering argument in order to go from a continuous hypothesis class to a discrete collection of functions.

The hardness result is conceptually based around a reduction to the poisoning coin problem from [Hanake et al 2022]. The main technical challenge is connected to handling information leakage: in a naive construction the behavior of non-target coins may reveal information about the target coin. This technical challenge is handled via sampling the biases of different coins independently from a certain distribution.

**Questions:**

- Suppose we want to establish high-probability bounds on the excess adversarial loss, rather than bounds on the expected adversarial loss. Does your work give such bounds?
- Are you aware of any work in ML security for which the poisoning budget is less than the inverse of the number of model parameters?
- Could you elaborate on similarities and differences of your work and [Bassily et al. 2019] and [Dagan and Feldman 2020]?

**Ethical Concerns:**

["NO or VERY MINOR ethics concerns only"]

**Final Justification:**

Good paper, but has some weaknesses potentially undermining the relevance of the paper. Authors note that the weaknesses could potentially be addressed, but leaves this for future work.

**Limitations:**

My understanding is that requiring the poisoning budget to be at most the inverse of the VC dimension could be a large limitation from the point of view of potential future impact on ML security. I think the work would benefit from discussing in more detail potential limitations due to this. Other than this, I think that the limitations are addressed.

**Quality:**

4

**Strengths And Weaknesses:**

Strengths
- The problem is fairly natural, and was studied in the previous work.
- The loss bounds are quite tight, because the work gives matching upper and lower bounds on excess risk up to logarithmic factors.
- The algorithmic approach based on exponential sampling could provide useful tools for future work. However, the paper does note that a similar approach was used in [Bassily et al. 2019] and [Dagan and Feldman 2020].

Weaknesses
- Arguably the more natural realizable setting was already solved in [Hanake et al 2022].
- I am not sure how much interest there within ML security community in methods for which the poisoning budget has to be less than $1/d$. This could potentially limit future impact of this work.

Some minor typos:
- Line 137 “as follow” should be “as follows”
- Between lines 137 and 138: “A target label y os drawn” -> “A target label y is drawn”

---

> ### Author Rebuttal · Authors · 2025-07-29
>
> We thank the reviewer for the detailed review and constructive suggestions. The questions raised are quite profound, or at least require substantial technical elaboration, so, unfortunately, we are not able to properly address them in the revision; however, we will flag them for future directions. Below, we give a detailed elaboration on them.
>
> - The reviewer comments that the realizable setting was already addressed in prior work [Hanneke et al. 2022].
>
> We note that the realizable and agnostic settings are complementary: each captures a different aspect of the learning landscape. While the realizable setting assumes the existence of a perfect concept that classifies all points correctly, the agnostic setting allows for arbitrary noise and aims to compete with the best possible hypothesis in a class of models. Both frameworks are well-established and widely studied in learning theory.
>
> One advantage of the agnostic setting is that it offers robustness to small deviations from ideal assumptions. In many practical scenarios, even when highly accurate classifiers exist, they may still incur some small but nonzero error—i.e., a nearly realizable setting that lies between the strict realizable and fully agnostic regimes. The agnostic framework captures this more realistic scenario and provides more resilient guarantees.
>
> This actually motivates further investigation into intermediate regimes such as near-realizability, which better reflect practical learning settings and offer a more nuanced understanding of robustness in the presence of noise.
>
> - Typos
>
> Fixed. Thanks.
>
> - Q.1. Do results generalize from in-expectation to high-probability?
>
> This is certainly a valid line of inquiry, and our gut feeling is that essentially the same bounds should hold also w.h.p., essentially because the exponential mechanism used in the learner that witnesses the upper bound is ``rather smooth''. However, exploring this direction, even formulating the relevant definitions, requires a fair amount of technical work. Thus, we do not address it in the present paper.
>
> - Q.2. Poisoning budget less than $1/d$, where $d$ is the number of parameters.
>
> We agree that for real-world models, the VC dimension can be extremely large. However, this reflects a well-known limitation of the classical PAC framework, which provides distribution-free guarantees and quantifies over all possible data distributions.
>
> In many practical settings, it is more appropriate to adopt data-dependent assumptions that make the learning problem more tractable and effectively decouple the VC dimension from the number of parameters. A canonical example is the analysis of support vector machines and kernel methods in infinite-dimensional spaces, where generalization depends not on the parameter count but on properties like the margin of the data. Such assumptions are conveniently formalized within the framework of PAC learnability of partial concept classes [1], which extends the classical PAC model to incorporate structure or distributional constraints.
>
> In this work, we focus on the basic PAC model to establish general upper and lower bounds. That said, our lower bound in Theorem 1 applies to partial concept classes as well, while the upper bound does not directly extend, as it relies on covering arguments. We will include this observation and suggest the extension of our results to partial concept classes as a direction for future work in the next draft.
>
> [1] A Theory of PAC Learnability of Partial Concept Classes. Noga Alon, Steve Hanneke, Ron Holzman, and Shay Moran. FOCS 2021
>
> - Q3. Similarities and differences with [Bassily et al. 2019] and [Dagan and Feldman 2020]
>
> Both mentioned papers study a different setup—namely, differentially private (DP) learning—but a parallel arises when designing learners that are resistant to poisoning attacks. Since a poisoner’s budget is limited, one strategy for achieving robustness is to ensure that the learner behaves similarly on similar datasets. This idea aligns with the stability guarantees required by DP, and so techniques from DP learning become relevant.
>
> The general approach of first constructing a finite $\varepsilon$-cover of the hypothesis class and then using it to learn is classical in statistical learning theory. In our setting, we use this approach in combination with the exponential mechanism to select a hypothesis. This template also appears in [Dagan and Feldman 2020] and [Bassily et al. 2019] in the context of differentially private (DP) learning, and we apply it here in the context of poisoning robustness. While the high-level structure is similar, there are several technical differences.
>
> In particular, our setting requires robustness not just to individual data point changes (as in standard DP), but to adversarial modifications of an entire $\eta$ fraction of the training set. This corresponds to what is known in the DP literature as group privacy, but in our case the group size is substantial. These strong stability requirements necessitate a different approach compared to [Dagan and Feldman 2020] and [Bassily et al. 2019]. Most notably, the proof and analysis of our algorithm are substantially different and also the parameters of the exponential mechanism are tuned differently due to our different stability requirement.
>
> Our analysis critically requires an $\varepsilon$-cover constructed using only $\tilde{O}(1/\varepsilon)$ examples—a realizability-type bound obtained by considering the class of symmetric differences of concepts—which is essential for achieving our excess error guarantees. In contrast, the more standard agnostic-type bound of $\tilde{O}(1/\varepsilon^2)$ (which follows directly from uniform convergence) would not suffice for our purposes. A similar reliance on the faster $\tilde{O}(1/\varepsilon)$ rate appears in [Bassily et al. 2019], where it is used to minimize the number of public data points required in their semi-private learning framework. Although the same covering property is used, our proof differs substantially due to the distinct objectives and stability requirements in our setting. We note that Dagan and Feldman [2020] also rely on the linear-in-$1/\varepsilon$ bound when constructing their cover; however, in their case, this yields only a quantitative improvement in privacy guarantees. In contrast, our analysis fundamentally hinges on this tighter bound—resorting to the more naïve quadratic $\tilde{O}(1/\varepsilon^2)$ rate would cause the argument to fail entirely.

---

> > ### Comment · Reviewer_yng4 · 2025-08-04
> >
> > Thank you for your response. I think it would be worthwhile to include into the paper the above comments on potential future directions, as well as the additional comparisons with previous work.
> >
> > My only additional comment is that the reference [1] mostly contains impossibility results for learning partial concept classes, even in the regular PAC learning framework. Then, it appears, the future extensions of this work to partial concept classes would have to focus on specific partial concept classes such as support vector machines under a margin assumption. It could potentially be useful for directing future work if authors included a list of such specific settings as open problems for future work (as opposed to only referencing [1], which mostly contains impossibility results).
> >
> > I think that it is most appropriate to keep my score at 4: Borderline accept.

---

### Official Review · Reviewer_fdBF · 2025-06-28

**Clarity:** 3
**Significance:** 3
**Originality:** 3
**Rating:** 5
**Confidence:** 4

**Summary:**

The paper studies the instance-targeted poisoning attack where the adversary can observe the target instance and the corrupt $\eta$ fraction of the training sample, with the goal of forcing the learner to make a mistake on the test instance. This problem has previously been studied under the realizable case and this paper extends it to the agnostic setting. A main result of the paper is the optimal excess error, i.e., the difference in error with the approximation error of the class for clean datasets, is $\Theta (\sqrt{\eta d})$, where $d$ is the VC dimension. The second interesting result is showing that private randomness does not help and the above excess rate error holds even when the randomness of the learner is public to the adversary. Moreover, the construction of the hard distribution for lower bound is not specific to the sample size as opposed to usual constructions in PAC learning. It is based on finding a finite set of hard distributions which proves that there is a fixed one that is hard for infinitely many samples sizes.

**Questions:**

It seems that the learner needs the knowledge of the poisoning budget. Is it clear (maybe from prior work) what happens if we want to relax such an assumption? Does the problem become significantly harder or are there straightforward techniques that can remove such a dependence with slight blowups in error rate?

Is it possible to consider a setting where the target is a test set rather than a single instance? The adversary then has the goal of making the learner to error on as many test instances as possible. I think we should know that the error rate in this case is still bounded by $O(\sqrt{\eta d})$. Is it possible to get better error rates? I am not much familiar with the setting of prior work and I am not sure how useful such a setting would be.

**Ethical Concerns:**

["NO or VERY MINOR ethics concerns only"]

**Final Justification:**

My initial assessment of the paper was positive. After reading rebuttals, I still keep my score. The authors explained well my questions and I agree that some of the questions are for future work. I raised them because the paper provokes them, which makes me believe the problem studied is an interesting research direction.

**Limitations:**

Yes.

**Paper Formatting Concerns:**

None.

**Quality:**

4

**Strengths And Weaknesses:**

I think generalizing this problem to the agnostic setting is a reasonable next step in this line of research and the result of the paper kind of answer the relevant questions under this setting in the agnostic case. I also think showing that private randomness cannot protect against such attacks is an interesting result of the paper. I still think the authors could have added more detail for other parts, e.g., the private vs. public randomness in the main paper. Maybe this is something that the authors can try for the camera-ready version. This is just a suggestion.

---

> ### Author Rebuttal · Authors · 2025-07-30
>
> We thank the reviewer for his or her excellent questions. Those are, to our view, some nice and interesting research directions, and we consider adding them as open questions (and acknowledge the anonymous reviewer for proposing them).
>
> We can hardly say anything on top of that regarding Question 1: How will the setup change if the learner does not know the poisoning budget? - It is indeed quite natural, but we have no idea of how to approach it at the moment.
>
> However, something can be said about Question 2: Is it possible to consider a setting where the target is a test set? - Standard uniform convergence arguments imply that any ERM on the (possibly poisoned) training sample will, on average, have test error at most $\sqrt{\frac{d}{m}} + \eta$ on a random test set of size $m$. Hence, when $m > \frac{1}{\eta}$, this bound becomes better than the worst-case bound we provide. More generally, it seems possible to interpolate between the case of a single test point (where the optimal error scales as $\sqrt{d\eta}$) and the case of a large test set (where the optimal value should approach $\eta$).

---

> > ### Comment · Reviewer_fdBF · 2025-08-09
> >
> > Thank you for your rebuttal. It seems interesting to me to think about different achievable rates for different target sizes. I am happy that the authors think the proposed questions are worth including in future work.

---

### Official Review · Reviewer_EBae · 2025-06-29

**Clarity:** 3
**Significance:** 3
**Originality:** 3
**Rating:** 5
**Confidence:** 2

**Summary:**

This paper addresses the *instance-targeted* poisoning problem, in which a Learner must effectively predict a test point in the presence of an adversary that can corrupt a fraction of the training examples. It specifically addresses a problem left open by Hanneke, Karbasi, Mahmoody, Mehalel, and Moran 2022: if there exists an (optimal) *randomized* learner that can learn in this instance-targeted poisoning setting in the general agnostic model of learning. The aforementioned paper of Hanneke, Karbasi, Mahmoody, Mehalel, and Moran 2022 address this problem in the *realizable* setting (proving optimal rates) and show the impossibility of a *deterministic* learner of achieving this in the agnostic case. This current paper's main result shows that, in fact, a *randomized* (proper) learner can achieve an optimal rate of $\Theta(\sqrt{d \eta})$ in the agnostic case of instance-targeted poisoning. The main results of the paper exhibit a lower bound $\Omega(\sqrt{d \eta})$ by reducing to what the authors call the "poisoned coin problem" and an upper bound by way of simply running exponential sampling over the hypothesis class $\mathcal{H}$.

**Questions:**

There are a couple of minor questions I have with some of the notation of the paper to address clarity:

1. On page 5: Step 2 of the "Poisoned Coin Problem with Oblivious Adversary" description is somewhat unclear to me. Why are you repeating the draw of the target label twice? Maybe this is just a typo.
2. On page 5: As a minor detail, in the description of the "$d$-Coin Problem," it would be helpful to explicitly include the adversary's choice of $p_1, \dots, p_d$ especially because this plays a key role in the "fix" of the construction. Also, I was wondering -- for the lifting to the VC dimension case, is it still the case that the lower bound is proven for an oblivious adversary? In the box for the "$d$-Coin Problem," it says that the adversary does indeed observe the training set and target example. So I was wondering if the formal reduction is to the oblivious case as well when we deal with $d$ coins?
3. If space permits in Section 2.2 for the Upper Bound, it might be helpful to formally state the anti-concentration property as a lemma as it is key to the argument.

As a general question:

4. It seems that the definitions of excess error in this setting given in Section 4 address in-expectation quantities. Do these same results port over to high probability? Or are there additional tools needed besides standard concentration arguments?

**Ethical Concerns:**

["NO or VERY MINOR ethics concerns only"]

**Final Justification:**

I maintain my original positive score but emphasize that my confidence is not as high as reviewers that are more familiar with the subfield.

**Limitations:**

Yes.

**Paper Formatting Concerns:**

I did not notice any formatting issues.

**Quality:**

3

**Strengths And Weaknesses:**

## Strengths

This paper's main results present a lower bound and upper bound for learning under instance-targeted poisoning in the agnostic setting, where an adversary has a general $\eta \in (0, 1)$ poisoning budget. It is generally well-written and clear, and it does address an existing open problem in the literature.

**Originality:** The paper seems, to my knowledge, original in the tools it uses to address an existing open problem. Although the authors note that the aforementioned Hanneke et al. paper in 2022 introduced the reduction to the poisoned coin problem, the authors adapt the lower bound to their setting and, interestingly, also show that the "hard distribution" is stronger than traditional PAC-style bounds in that the adversary has a single fixed distribution instead of a distribution tailored to $n$ (the sample size). They do this by essentially allowing the adversary to construct the hard distribution (which is a set of biases $p_1, \dots, p_d$ for $d$ coins) not by fixing these biases deterministically, but, rather, letting the adversary *draw* the biases (in a somewhat "Bayesian prior" way) over a fixed distribution over $[0, 1]$. For the upper bound, the author's techniques must be original because the prior work in this subfield established that it is impossible to achieve an upper bound for this setting with a deterministic learner. Indeed, the authors show that, with a specific exponential sampling mechanism, they can achieve the upper bound as well.

**Quality:** The paper's claims seem to be technically correct, at least when judging the proof sketches. One issue that I had was further intuition and discussion towards how the constructions the authors introduce might be relevant in a defense to a real-world adversarial poisoning attack, but, because this is a theory paper, that would just be a nice cosmetic addition.

**Clarity:** The authors provide clear proof sketches and lay out the paper's results in an intuitive and organized way. In particular, the main theorems were nicely outlined and it was mostly clear in the lower bound exposition what the "hard" problem was and what the main steps to developing the algorithm was in the upper bound section. There are a couple of small confusing possible typos that I found in some notation, which is outlined in the **Questions** section.

**Significance:** The work fills in an existing gap in our understanding of the instance-targeted poisoning problem, particularly in the agnostic case. With that, it does round out the entire basic understanding of this setting, as sharp results for the realizable case have also been shown. I am not entirely familiar with this line of literature (and the broader poisoning attack literature in general), so I cannot be a judge to whether this particular *subfield* is significant in the broader research agenda in adversarial ML. However, as a specific learning theory paper addressing a specific problem, it does address and close an open question.

## Weaknesses
One might point that the main possible weaknesses of the paper might be that the problem's solutions don't introduce any new technical techniques. The lower bound appeals to an existing hard problem (the poisoned coin problem) but it wasn't entirely clear to me what the new technical additions were needed to address the agnostic case. It would be helpful to include a sentence or two about this in the lower bound section to make clear how this hard distribution differs from the realizable construction. For the upper bound, the algorithm itself is not terribly complex, though I don't think that's necessarily a knock on the paper -- if a simple algorithm suffices, then a simple algorithm suffices. This all being said, the paper clearly sets out to address a known open problem and rounds out our understanding of a learning-theoretic setting, so I believe how "technical" the means to those ends are secondary to the fact that these results are sharp.

---

> ### Author Rebuttal · Authors · 2025-07-29
>
> We thank the reviewer for a thorough elaboration on the paper and for his or her constructive remarks. Here's our address of the points raised:
>
> - Weakness: How does the coin problem get adapted to the setting of the paper? - While the coin problem was indeed introduced in Hanneke et.al., and even in the context of an agnostic setting, the learner there was assumed to be deterministic. Allowing it to be randomized changes the problem quite substantially, both in terms of the result and the technique needed to obtain it. We added a note pointing this out in the paper.
>
> - Q.1. Typo in Step 2 of the description of the ``Poisoned coin with Oblivious Adversary.'' - Yes, this was a typo, fixed.
>
> - Q.2. Description of the $d$-Coin Problem. - First of all, as suggested, we've emphasized the adversarial choice of $p_1, \dots, p_d$. Then, you are right - the lower bound in the general case is indeed proven via oblivious adversary, and we added the respective note to the paper text.
>
> - Q.3. Anti-concentration bounds. - Sorry for the confusion. While anti-concentration bounds can indeed be used to prove the upper bound for the toy 1-coin problem, we failed to adapt this approach to the real problem, and used instead the exponential mechanism from differential privacy. Thus, the anti-concentration is not used, and is mentioned only as an analogy. We clarified this in the text, and, because of that, and also due to space restrictions, we are not adding their exact formulation. On a private note, for 1-coin, one can use, for example, Khintchine's for $p=1$, that is, that the sum of $n$ Bernoullis is w.h.p. at distance ~ $O(\sqrt{n})$ from their expected value.
>
> - Q.4. Do results generalize from in-expectation to high-probability? - This is an interesting question, and our gut feeling is that essentially the same bounds should hold also w.h.p., essentially because the exponential mechanism used in the learner that witnesses the upper bound is ``rather smooth''. However, exploring this direction, even formulating the relevant definitions, requires a fair amount of technical work. Thus, we will not address it in the present paper.

---

> > ### Comment · Reviewer_EBae · 2025-08-06
> >
> > Thank you for acknowledging my comments and for your proposed adjustments to the paper! I will keep my score.

---

### Official Review · Reviewer_Yaf2 · 2025-07-03

**Clarity:** 3
**Significance:** 2
**Originality:** 3
**Rating:** 4
**Confidence:** 2

**Summary:**

This paper follows with the Hanneke et al. (NeurIPS 2022), study the problem of learning with an adversary that can corrupt $\eta$ fraction of the training examples in order to cause error on a given point.Author gives an upper and a lower bound about $O(\sqrt{d\eta})$ for excess error for random algorithm. Then, for the Private and Public poison problem, author shows that under public, we can guarantee the robustness as good as private.  In Technical Overview, author shows how to use coin problem to solve their theorem.

**Questions:**

Question:

1:How is the target of attack (cause error on a given point) reflected in the theorem 1? The author mainly analysis the excess error.

2: In th2, if we take $A_{priv}$ as SGD, what is $A_{pub}$?

3: How to verify th1 in experiment?

**Ethical Concerns:**

["NO or VERY MINOR ethics concerns only"]

**Final Justification:**

The author has solved most of my questions.

**Limitations:**

no.

**Paper Formatting Concerns:**

no.

**Quality:**

3

**Strengths And Weaknesses:**

Strong:
(1): This paper is easy to read.The theory is novel.

Weakness:
I am not particularly familiar with this field, but some parts of the theorem do confuse me.(If author can well answer, I will improve my score)

(1)	:The boundaries given in the Th1 seem too loose in the real world, because VC-dim d is always much large than poison rate $1/\eta$, such as poison $1\%$ for ResNet trained on CIFAR-10.

(2):Theorem 2 only compared the robustness of two algorithms(Private and Public) in poisoning, but did not analyze their accuracy on clean data. As is well known, improving robustness may lead to a decrease in generalization, but the author did not discuss this point


(3): In Theorem 3, the author states that for an infinite number of n samples, the excess-error will be $O(\sqrt{d\eta})$, but does not mention the probability of sampling such n samples, which should be mentioned. On the other hand, designing distributions and attack methods based on training set size $n$ seems to be impractical in the real world, what is the importance of th3?

---

> ### Author Rebuttal · Authors · 2025-07-30
>
> We thank the reviewer for taking the time to read our paper, reflect on its contributions, and share constructive comments. As detailed below, we will take these suggestions into account in the next revision of the manuscript.
>
> ⸻
>
> **(1) Concern about the VC dimension in Theorem 1 being too large in practice.**
>
> We agree that for real-world models such as ResNet trained on CIFAR-10, the VC dimension can be extremely large. However, this reflects a well-known limitation of the classical PAC framework, which provides distribution-free guarantees and quantifies over all possible data distributions.
>
> In many practical settings, it is more appropriate to adopt data-dependent assumptions that make the learning problem more tractable and effectively decouple the VC dimension from the number of parameters. A canonical example is the analysis of support vector machines and kernel methods in infinite-dimensional spaces, where generalization depends not on the parameter count but on properties like the margin of the data. Such assumptions are conveniently formalized within the framework of PAC learnability of partial concept classes [1], which extends the classical PAC model to incorporate structure or distributional constraints.
>
> In this work, we focus on the basic PAC model to establish general upper and lower bounds. That said, our lower bound in Theorem 1 applies to partial concept classes as well, while the upper bound does not directly extend, as it relies on covering arguments. We will include this observation and suggest the extension of our results to partial concept classes as a direction for future work in the next draft.
>
> [1] A Theory of PAC Learnability of Partial Concept Classes. Noga Alon, Steve Hanneke, Ron Holzman, and Shay Moran. FOCS 2021
>
> ⸻
>
> **(2) Concern that Theorem 2 does not analyze performance on clean data.**
>
> Theorem 2 applies for any poisoning rate η, including η = 0, which corresponds to clean data. In this case, the result becomes somewhat trivial since the adversary has no influence over the training data. Still, the guarantee holds: the algorithm with public randomness performs no worse (in terms of excess loss) than the one using private randomness.
>
> ⸻
>
> **(3) Concern about the interpretation and importance of Theorem 3.**
>
> The goal of Theorem 3 is to investigate whether stronger guarantees on excess loss can be achieved by moving from the worst-case, distribution-free PAC framework to the more fine-grained, distribution-dependent setting of universal learning [2].
>
> In the absence of data poisoning (i.e., η = 0), it is well known that PAC learning rates scale as VC/n in the realizable case and as √(VC/n) in the agnostic case. In contrast, in the universal learning framework—where performance is measured pointwise over distributions—many hypothesis classes (including every online learnable class) enjoy significantly faster rates, often with exponentially fast decay of the learning curve.
>
> This raises the natural question: can one beat the √(dη) bound from Theorem 1 by moving to the universal setting? Theorem 3 shows that the answer is negative—despite the extra granularity of universal learning, the same asymptotic √(dη) limitation persists. We will clarify this motivation in the revised text.
>
> [2] A Theory of Universal Learning. Olivier Bousquet, Steve Hanneke, Shay Moran, Ramon van Handel, and Amir Yehudayoff. STOC 2021
>
> ⸻
>
> **Responses to Additional Questions:**
>
> - Q1: How is the target of attack (cause error on a given point) reflected in Theorem 1?
>
> Theorem 1 guarantees that for any learning algorithm, the adversary we construct can force an excess loss of at least roughly √(dη), where d is the VC dimension and η is the poisoning rate. This excess loss is uniform across all algorithms.
>
> However, the absolute error rate on the target point (i.e., the total loss) depends on the specific distribution chosen by the adversary, which in turn may depend on the algorithm being attacked. For some algorithms, the total loss on the target point may approach 0.5; for others, it may be significantly lower. We will clarify this point in the revised version.
>
> ⸻
>
> - Q2: In Theorem 2, if we take the learner to be SGD, what is the corresponding public learner?
>
> To implement the public learner, it suffices to estimate the probability P that the private learner predicts label 1 on the given test point x. Once P is estimated, the public learner draws a random number R uniformly from [0,1]; if R < P, it outputs label 1, and otherwise it outputs 0. This simulates the private learner’s behavior using only public randomness.
>
> ⸻
>
> - Q3: How to verify Theorem 1 experimentally?
>
> Theorem 1 is an information-theoretic result that quantifies over all distributions, all adversaries (in the upper bound), and all learning algorithms (in the lower bound). As such, it is not meant to be verified empirically—doing so would require checking infinitely many cases, even for simple hypothesis classes. That said, the mathematical proof certifies the validity and tightness of the bounds.
>
> Empirical exploration might still be meaningful for fixed distributions and concrete attack strategies relevant to real-world applications, but such experiments lie beyond the scope of this theoretical work.

---

> > ### Comment · Reviewer_Yaf2 · 2025-08-06
> >
> > The author has solved most of my questions. So I will improve my score.

---

### Decision · Program_Chairs · 2025-09-17

**Decision:**

Accept (spotlight)

**Comment:**

This paper delivers ​exceptional theoretical contributions: resolving an open problem with tight bounds, introducing algorithmic innovations, and redefining assumptions about randomness in adversarial settings. While practical deployment faces inherent PAC limitations (e.g., large d), the work provides foundational insights that will influence multiple ML subfields. The authors’ rigorous engagement with reviewers solidified unanimous support. I endorse ​acceptance​.